# Factors influencing the participation of pregnant and lactating women in clinical trials: A mixed-methods systematic review

Mridula Shankar[1]☯*, Alya Hazfiarini[1]☯, Rana Islamiah Zahroh[1], Joshua P. Vogel[2], Annie R. A. McDougall[2], Patrick Condron[3], Shivaprasad S. Goudar[4], Yeshita V. Pujar[4], Manjunath S. Somannavar[4], Umesh Charantimath[4], Anne Ammerdorffer[5], Sara Rushwan[5], A. Metin Gülmezoglu[5], Meghan A. Bohren[1]

1 Gender and Women's Health Unit, Nossal Institute for Global Health, School of Population and Global Health, University of Melbourne, Carlton, Victoria, Australia, 2 Maternal, Child and Adolescent Health Program, Burnet Institute, Melbourne, Victoria, Australia, 3 University Library, University of Melbourne, Carlton, Victoria, Australia, 4 Women's and Children's Health Research Unit, KLE Academy of Higher Education and Research, Jawaharlal Nehru Medical College, Belagavi, Karnataka, India, 5 Concept Foundation, Geneva, Switzerland/Bangkok, Thailand

☯ These authors contributed equally to this work.
* mridula.shankar@unimelb.edu.au

**Data Availability Statement:** All relevant data are within the manuscript and its Supporting Information files.

**Funding:** The research in this publication was supported by funding from MSD (grant MFM-22-

## Abstract

### Background

Poor representation of pregnant and lactating women and people in clinical trials has marginalised their health concerns and denied the maternal–fetal/infant dyad benefits of innovation in therapeutic research and development. This mixed-methods systematic review synthesised factors affecting the participation of pregnant and lactating women in clinical trials, across all levels of the research ecosystem.

### Methods and findings

We searched 8 databases from inception to 14 February 2024 to identify qualitative, quantitative, and mixed-methods studies that described factors affecting participation of pregnant and lactating women in vaccine and therapeutic clinical trials in any setting. We used thematic synthesis to analyse the qualitative literature and assessed confidence in each qualitative review finding using the GRADE-CERQual approach. We compared quantitative data against the thematic synthesis findings to assess areas of convergence or divergence. We mapped review findings to the Theoretical Domains Framework (TDF) and Capability, Opportunity, and Motivation Model of Behaviour (COM-B) to inform future development of behaviour change strategies.

We included 60 papers from 27 countries. We grouped 24 review findings under 5 overarching themes: (a) interplay between perceived risks and benefits of participation in women's decision-making; (b) engagement between women and the medical and research ecosystems; (c) gender norms and decision-making autonomy; (d) factors affecting clinical trial recruitment; and (e) upstream factors in the research ecosystem. Women's willingness

159697 to Concept Foundation), through its MSD for Mothers initiative (https://www.msdformothers.com/) and is the sole responsibility of the authors. MSD for Mothers is an initiative of Merck & Co., Inc., Rahway, NJ, U.S.A. MAB's time is supported by an Australian Research Council Discovery Early Career Researcher Award (DE200100264) and a Dame Kate Campbell Fellowship (University of Melbourne Faculty of Medicine, Dentistry and Health Sciences). JPV is supported by an Australian National Health and Medical Research Council (NHMRC) Investigator grant (GNT1194248). The funders had no role in the study design, data collection and analysis, decision to publish, or preparation of the manuscript.

**Competing interests:** The authors have declared that no competing interests exist.

**Abbreviations:** BCW, Behaviour Change Wheel; COM-B, Capability, Opportunity, and Motivation Model of Behaviour; MMAT, Mixed Methods Appraisal Tool; TDF, Theoretical Domains Framework.

to participate in trials was affected by: perceived risk of the health condition weighed against an intervention's risks and benefits, therapeutic optimism, intervention acceptability, expectations of receiving higher quality care in a trial, altruistic motivations, intimate relationship dynamics, and power and trust in medicine and research. Health workers supported women's participation in trials when they perceived clinical equipoise, had hope for novel therapeutic applications, and were convinced an intervention was safe. For research staff, developing reciprocal relationships with health workers, having access to resources for trial implementation, ensuring the trial was visible to potential participants and health workers, implementing a woman-centred approach when communicating with potential participants, and emotional orientations towards the trial were factors perceived to affect recruitment. For study investigators and ethics committees, the complexities and subjectivities in risk assessments and trial design, and limited funding of such trials contributed to their reluctance in leading and approving such trials. All included studies focused on factors affecting participation of cisgender pregnant women in clinical trials; future research should consider other pregnancy-capable populations, including transgender and nonbinary people.

## Conclusions

This systematic review highlights diverse factors across multiple levels and stakeholders affecting the participation of pregnant and lactating women in clinical trials. By linking identified factors to frameworks of behaviour change, we have developed theoretically informed strategies that can help optimise pregnant and lactating women's engagement, participation, and trust in such trials.

## Author summary

### Why was this study done?

- Pregnant and lactating women and people are routinely excluded from participating in drug and vaccine clinical trials, resulting in limited options for prevention and treatment of medical conditions.

- Challenges to including pregnant and lactating women and people in clinical research have been identified at multiple levels of the research and health systems, but the full range of barriers and facilitators to participation are not well known.

### What did the researchers do and find?

- We conducted a mixed-methods systematic review and identified 60 research articles from 27 countries on the views and experiences of pregnant and lactating women's participation in clinical research, from the perspectives of cisgender women, family and community members, health workers, and people involved in the conduct of clinical research.

- Using a thematic synthesis approach, we identified barriers affecting participation including women having a limited appetite for risk during pregnancy and lactation, concerns about women's bodily autonomy during pregnancy, and challenges in obtaining ethical approval for clinical research with pregnant women.

- We also identified facilitators of participation including the potential for personal health benefits, expectations of higher quality care, trust in the medical and research systems, and strong teamwork between researchers and health workers.

### What do these findings mean?

- Our findings demonstrate the need for multipronged strategies to address barriers and reinforce facilitators across the various levels of the research and health systems.

- The actions that are needed to overcome these barriers and reinforce facilitators must be discussed, prioritised, and adapted to specific contexts.

- All included studies focused on factors affecting participation of cisgender pregnant women in clinical trials; future research should consider other pregnancy-capable populations, including transgender and nonbinary people.

## Introduction

Clinical trials are the foundation for knowledge on the efficacy and safety of biomedical interventions to protect health and treat illness. The fundamental questions of who participates and whose data contributes to trials have implications for understanding the risks and benefits of interventions, and the societal value of such interventions to specific populations. Pregnant and lactating women and people have long been underrepresented or excluded entirely from participating in therapeutic and vaccine clinical trials [1]. Notwithstanding valid concerns regarding fetal and infant safety, an outright exclusionary response to this complex issue has denied the maternal–fetal/infant dyad the health benefits of biomedical innovation, despite demonstrated public health need [2,3]. As a recent example, during the COVID-19 pandemic, pregnant women and people were excluded from early therapeutic and vaccine trials despite greater severity of infection-related illness [4–9].

Including pregnant and lactating women and people as research participants is vital: pregnancy is a unique physiological state where the body undergoes adaptations that can lead to pregnancy-specific disorders or worsen preexisting conditions [10]. These changes can influence how effective a drug is, whether and how the body responds to the drug, and the dosages at which the drug is optimally effective and minimally harmful. Most pregnant women take at least 1 medication during pregnancy [11], yet many of these medications are provided with limited information on efficacy, appropriate dosing, and safety in these populations [1]. Pregnant and lactating women with preexisting illnesses may also be advised to discontinue medications to minimise potential harms, without full appreciation of the possible consequences of unmedicated disease progression [12].

The current state of maternal health and the limited therapeutic options available for pregnant and lactating populations illustrates the consequences of these evidence gaps. Each year, complications of pregnancy and childbirth result in approximately 287,000 maternal deaths

[13], 1.9 million stillbirths [14], and 2.3 million neonatal deaths [15]. Most of these deaths occur from preventable or treatable obstetric causes (e.g., postpartum haemorrhage, pre-eclampsia/eclampsia, sepsis) that are generally treated using repurposed medications that were originally developed and approved for use in other non-obstetric conditions [16]. Over the past 3 decades, only 2 drugs have been registered to specifically treat pregnancy-related complications: Atosiban—a tocolytic to prevent preterm birth, and Carbetocin—an oxytocin analogue for managing postpartum haemorrhage [17]. Pregnancy-specific medicines rarely progress through the research and development pipeline due to a multitude of factors, including the absence of public stewardship, chronic underinvestment, and regulatory and market barriers [18,19]. Maternal mortality rates have largely remained static in the Sustainable Development Goal era: progress has halted or reversed in 150 countries [13]. Without significant investments in pharmaceutical development, the 2030 target of a global maternal mortality ratio less than 70 maternal deaths per 100,000 live births [20] is unlikely to be achieved.

Poor representation of pregnant and lactating women and people in clinical research, and the absence of a pregnancy-focused research and development agenda violates fundamental ethical principles of justice and equity [12,21]. Challenges to equitable inclusion operate across all research stages: "upstream" barriers include a lack of appropriate animal models, pharmaceutical industry risk aversion, and clinical trials and liability insurance challenges [12,18,22,23]. "Downstream" barriers include perceptions that pregnant and lactating women do not want to take part in clinical trials, or that their inclusion makes research activities too risky or onerous [23]. Overall, there is a lack of a comprehensive understanding of the full range of these factors from the perspectives of key stakeholder groups. This mixed-methods systematic review seeks to address this gap by synthesising current research evidence on factors (i.e., barriers and facilitators) affecting the participation of pregnant and lactating women in vaccine and therapeutic clinical trials. We use behavioural [24,25] frameworks to provide a theory-informed basis for the development and implementation of appropriate behaviour change intervention strategies to promote their meaningful inclusion.

## Methods

This review is reported using the Preferred Reporting Items for Systematic Reviews and Meta-Analyses (PRISMA) guidelines (S1 Appendix), Enhancing Transparency in Reporting the Synthesis of Qualitative Research (ENTREQ) statement (S2 Appendix), and based on guidance from the Cochrane Effective Practice and Organisation of Care group [26]. The protocol has been registered (PROSPERO: CRD42023462449).

### Types of studies

We included primary qualitative, quantitative, and mixed-methods studies. There were no limitations on publication date, language, or country.

We excluded publications that were not primary research, including conceptual scholarship on the ethics of inclusion/exclusion, case reports, reviews, commentaries, short communications, editorials, news articles, letters to the editor, conference abstracts, workshop summaries, theses or dissertations, book chapters, book reviews, and regulatory or committee guidance or decisions.

### Topic of interest

This review focuses on systematically identifying the factors, including barriers and facilitators, influencing the participation of pregnant and lactating women in drug or vaccine trials (i.e., therapeutic or prophylactic trials). We recognise that people who are capable of pregnancy

have diverse gender identities. We use the terminology "pregnant and lactating women," acknowledging that empirical literature on this topic has been focused on the experiences of cisgender women. Extrapolating these data to apply to people with other gender identities may lead to inaccurate or incomplete conclusions.

We included studies that described the attitudes, perspectives, and experiences of multiple stakeholders: women who participated and declined participation in clinical trials during pregnancy and lactation, partners or husbands, family members, community leaders, health workers, research staff, study investigators, ethics committee members, regulators, funders, pharmaceutical representatives, policy makers, and other relevant stakeholders.

We excluded the following types of interventions from this review: (a) lifestyle or behavioural interventions; (b) trials of diagnostics or medical devices; (b) workforce interventions to improve clinical care outcomes; (c) alternative or complementary medicine; (d) trials evaluating health policies or clinical protocols; (e) fetal tissue research, bio-banking, and genetic testing; (f) facilitators and barriers to engaging pregnant women in observational research; (g) supports to clinicians or pregnant or lactating women regarding decision-making on medication; and (h) research solely focused on substance use prevention and treatment, due to the particularly distinct barriers and facilitators given overlapping vulnerabilities among substance-using pregnant women, and unique considerations in relation to fetal health such as in utero exposure to alcohol and other substances. We also excluded clinical trial protocols and publications of randomised controlled trials that did not contain data related to facilitators or barriers to trial participation.

## Search methods for identification of relevant studies

We searched 8 databases from inception to 14 February 2024: MEDLINE (Ovid), CINAHL Complete, Family & Society Studies Worldwide, SocINDEX, Scopus, Web of Science Core Collection, Embase (Ovid), and Global Health (Ovid). PC, an Information Specialist developed the final search strategy (S3 Appendix), using a combination of terms relevant to pregnant and lactating women, and perspectives and experiences of stakeholders regarding their inclusion/exclusion and participation in drug or vaccine clinical trials. No restrictions were placed on publication year, language, or geographical setting.

## Selection of studies

We imported the search results into Covidence [27] and removed duplicates. Five review authors (MS, AH, MAB, AM, and AA) independently screened titles and abstracts. Titles and abstracts of non-English publications were screened with the assistance of Google Translate. Three reviewers (MS, AH, and AM) independently reviewed full texts. One French publication that met the inclusion criteria was translated to English using ChatGPT [28], and translation accuracy was subsequently verified with a native French speaker in our research network. At each screening stage, differences in decisions regarding record inclusion were resolved through discussion and final decisions were made through consensus with a third review author (MAB).

## Data extraction and assessing methodological limitations

Two review authors (MS and AH) extracted relevant data, including study aims, methodological characteristics, geographical settings, population of interest (pregnant women, lactating women, or both), intervention type (therapy or vaccine), specific areas of research, and study findings (author-generated themes, supporting explanations, participant quotes, survey results, and relevant tables and figures). We developed a data extraction form and refined it by

extracting data from a subset of 6 studies. All extracted data was cross-checked for accuracy and completeness, and differences resolved via consensus.

Two reviewers (MS and AH) independently assessed the methodological limitations of each study using an adapted Mixed Methods Appraisal Tool (MMAT) [29]. For qualitative studies, evaluative criteria included alignment of methodology and data collection with research aims, rigour in data analysis and reporting of study findings, ethical considerations, and researcher reflexivity. We assessed quantitative studies based on the suitability of sampling strategy, reporting on sample representativeness, use of appropriate measures, level of nonresponse bias, ethical considerations, and relevance of statistical analyses conducted. In addition to the aforementioned criteria, we assessed mixed-methods studies to determine whether authors demonstrated sufficient rationale for the use of a mixed-methods approach, effectiveness of integration of study components and outputs, and discussion of data triangulation. All differences in assessments between the 2 review authors were resolved through discussion. The assessment of methodological limitations did not affect the inclusion or exclusion of studies but rather served as a mechanism for determining confidence in the evidence.

## Data analysis and synthesis

We used a thematic synthesis approach to analyse qualitative data [30]. After selecting 6 data-rich studies, 2 reviewers (MS and AH) independently applied line-by-line coding to the textual data to create summative codes. Codes were discussed for consistency in meaning and refined if necessary. The remaining studies were each coded by one of the 2 reviewers, and new codes were added as necessary. Through discussion, we subsumed codes of similar meaning under broader categories, gradually developing "summary layers" in a hierarchical grouping structure. We applied the gender domains of the gender analysis matrix [31] as a lens to our findings to understand how our data on factors influencing participation were shaped by aspects such as distribution of labour and roles, gender norms and beliefs, access to resources, decision-making power, and institutional policies. We consolidated our results into a set of 5 overarching themes and 24 review findings through an iterative process of identifying, comparing, and discussing conceptual boundaries between and among thematic data outputs.

Two review authors (MS and AH) used the GRADE-CERQual (Confidence in the Evidence from Reviews of Qualitative research) approach [32,33] to assess our confidence in each of the 24 qualitative review findings. GRADE-CERQual assesses confidence in the evidence, based on the following 4 key components [26]:

1. methodological limitations of included studies [34];

2. coherence of the review finding [35];

3. adequacy of the data contributing to the review finding [36]; and

4. relevance of the included studies to the review question [37].

After assessing each component, we made a judgement via consensus about the overall confidence—rated as high, moderate, low, or very low—in the evidence supporting the review finding [32]. Detailed descriptions of the GRADE-CERQual assessments are in S4 Appendix.

We then mapped data from the quantitative studies onto the findings of the qualitative evidence synthesis, and determined areas of convergence or divergence, and whether any additional factors arose that had previously not been discussed. We regarded the quantitative data as (a) "supporting" of a qualitative evidence synthesis finding if the information synthesised from the contributory quantitative studies were similar to the finding; (b) "extending" if the data offered additional details in line with a review finding; and (c) "contradictory" if the data

conflicted with a review finding. Summaries of the quantitative findings are presented in
S5 Appendix.

Finally, we mapped our review findings to the Theoretical Domains Framework (TDF) [24]
and the Capability, Opportunity, and Motivation (COM-B) [25] models of behavioural deter-
minants and the Behaviour Change Wheel (BCW) to identify and provide a rational basis for
the development and implementation of appropriate behaviour change strategies.

### Review team and reflexivity

The review author team has diverse personal backgrounds, including gender, personal experi-
ences of pregnancy, countries of origin and residence, and linguistic traditions. Our profes-
sional and academic backgrounds and experiences are varied, and include the social,
behavioural, and biomedical sciences, medicine, clinical epidemiology, and public health.
Some review authors have led and implemented trials in maternal and perinatal health. As an
interdisciplinary team with diverse social and professional backgrounds, we maintained a
reflexive stance through all stages of the review process by engaging in multiple reflective dia-
logues to interrogate and interpret the data and findings. Through this process, we named and
critiqued assumptions that underpinned the analysis and challenged disciplinary biases. In
doing so, we aimed to develop review findings that were inclusive of different disciplinary
lenses.

## Results

Sixty papers from 53 studies met the inclusion criteria [38–97]. Fig 1 presents the PRISMA
flowchart. Table 1 reports the summary characteristics of included papers and S6 Appendix
includes more detailed individual characteristics of the included papers.

### Description of papers

Thirty-nine papers used qualitative methodologies [39,40,42–48,53,54,56–66,69,70,72–
74,78,81,82,84–87,89–92,96], 18 papers used quantitative methodologies [38,41,50–
52,67,68,71,75–77,79,80,88,93–95,97], and 3 papers used mixed-methods study designs
[49,55,83].

The 60 papers present data from 27 countries and 4 geographic regions: 13 countries in
**Africa** [44–47,65,73,78,84,85], 8 countries in **Europe** [38,39,41,48–50,53–
56,58,59,61,62,64,67–69,72,74,80–83,86,89,90,92,94,96], 3 countries in **the Americas**
[42,43,51,52,57,60,63,66,70,71,75,77,79,85,88,91,93,95], and 3 countries in the **Western Pacific**
[40,76,87,97].

Fifty-one papers focused on pregnant women only [38–41,44,47–50,52,53,55–70,72–94,97],
2 papers focused on lactating women only [46,96], and 7 papers focused on pregnant and lac-
tating women [42,43,45,51,54,71,95]. Thirty-seven papers addressed a therapeutic drug-related
intervention [38,40,41,44–49,53,56,59–62,66,69,70,72,73,77,79–90,92,93,96,97], 11 papers
focused on a vaccine-related intervention [50,51,55,57,58,63,64,67,68,78,94], and 12 papers
were about pregnant and/or lactating women's participation in interventional clinical trials
generally [39,42,43,52,54,65,71,74–76,91,95].

Twenty-five papers included perspectives of pregnant women
[38,45,47,48,51,57,58,60,61,64,65,67,71–75,77,85,89–91,94,95,97], 28 papers included perspec-
tives of postpartum women [39–41,44–46,49,51,56,57,59,62,63,69–71,74,79–87,92,95], and 14
papers included health workers' perspectives [44,47,50,52–54,61,64,65,67,87,88,91,94]. For
other stakeholder groups, please refer to Table 1.

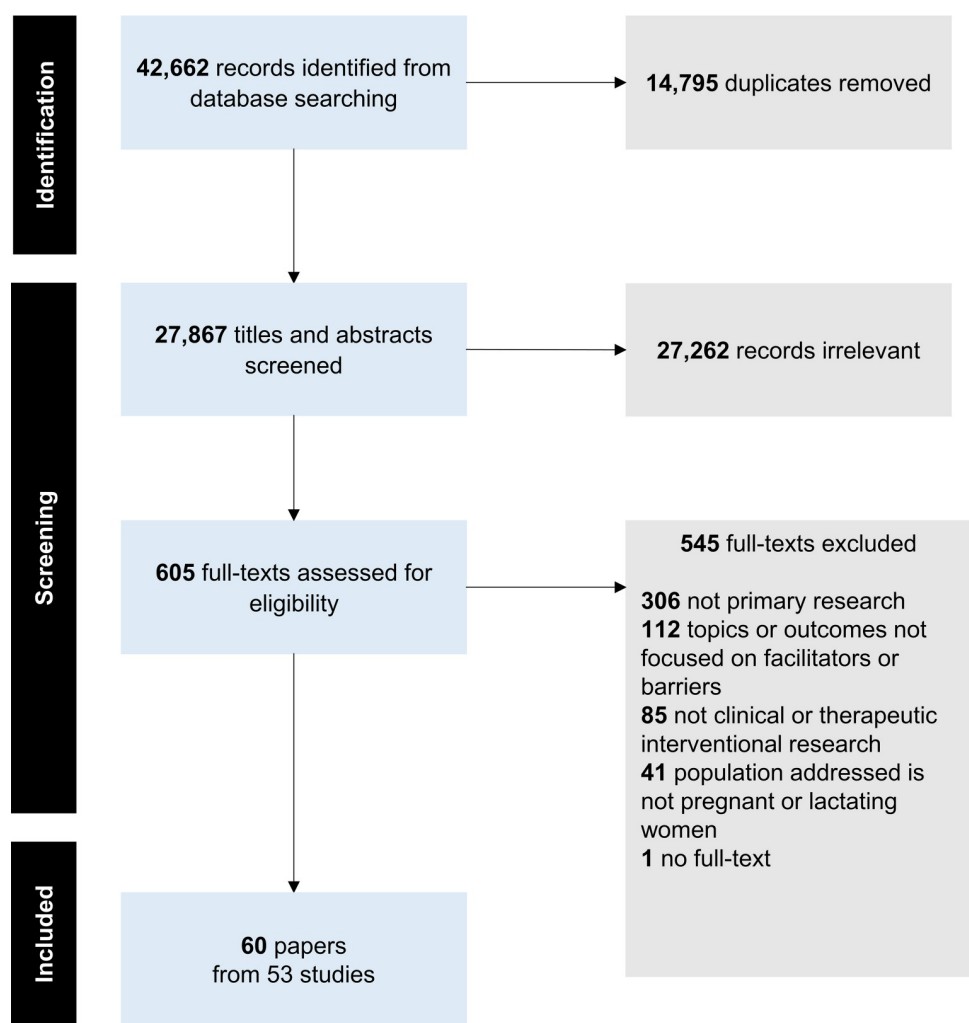

**Fig 1. PRISMA flowchart depicting search and selection process.**

## Methodological limitations of included studies

Assessments of methodological limitations of the included studies are available in S7 Appendix. Across qualitative studies, the most common methodological limitations concerned recruitment approaches and strategies, descriptions of analytical methods, ethical considerations, specifically steps or precautions taken to protect from loss of privacy and confidentiality, data security and integrity, and most studies did not include a reflexivity statement. Across quantitative studies, authors rarely reported on indicators of sample representativeness of the target population, most did not report on or were judged at high risk of nonresponse bias, and ethical considerations pertaining to data security and integrity were frequently missing. For the 3 mixed-methods studies, limitations were identified at the level of integrating methodological approaches at the methods, interpretation, and reporting levels.

## Themes and findings from the qualitative and quantitative evidence synthesis

We developed 5 overarching themes and 24 review findings in the qualitative evidence synthesis (Table 2):

**Table 1. Summary of characteristics of included papers.**

| Characteristics | n = 60 | % |
|---|---|---|
| **Research methods** | | |
| Qualitative | 39 | 65.0 |
| Quantitative | 18 | 30.0 |
| Mixed-methods | 3 | 5.0 |
| **Study region (n = 61)[a]** | | |
| Africa | 9 | 14.8 |
| Europe | 30 | 49.2 |
| The Americas | 18 | 29.5 |
| Western Pacific | 4 | 6.5 |
| **Country income level (n = 63)[b]** | | |
| High-income | 50 | 79.4 |
| Middle-income | 6 | 9.5 |
| Low-income | 7 | 11.1 |
| **Focal group** | | |
| Pregnant women | 51 | 85.0 |
| Lactating women | 2 | 3.3 |
| Pregnant and lactating women | 7 | 11.7 |
| **Intervention type** | | |
| Therapeutic-related intervention | 37 | 61.7 |
| Vaccine-related intervention | 11 | 18.3 |
| Interventional trial: type unspecified | 12 | 20.0 |
| **Trial area (n = 68)[c]** | | |
| Pregnancy/childbirth complications | 23 | 33.8 |
| Fetal/newborn research | 7 | 10.3 |
| Infectious diseases | 23 | 33.8 |
| Noncommunicable diseases | 2 | 2.9 |
| Mental health | 2 | 2.9 |
| Trial areas unspecified | 11 | 16.2 |
| **Stakeholder groups (n = 102)[d]** | | |
| Pregnant women | 25 | 24.5 |
| Postpartum women | 28 | 27.5 |
| Lactating women | 1 | 1.0 |
| Women who had breastfed in previous 5 years | 1 | 1.0 |
| Reproductive-aged women | 1 | 1.0 |
| Partners/husbands | 4 | 3.9 |
| Other family members | 1 | 1.0 |
| Community leaders and members | 3 | 2.9 |
| Health workers | 14 | 13.6 |
| Research staff | 9 | 8.8 |
| Research investigators | 7 | 6.9 |
| Ethics committee members, administrators, and regulators | 7 | 6.9 |
| Funders | 1 | 1.0 |

[a] One study conducted in Malawi (Africa) and United States (The Americas) [85].

[b] One study conducted in low- and high-income countries [85], and 1 study conducted in low- and middle-income countries [78].

[c] Eight studies address 2 trial areas [40,41,45,46,48,60,85,94].

[d] Twenty-nine papers reported more than 1 participant group [43–45,47,51,53–55,57,61,62,64–67,71,73,74,82,85–87,89–91,94–97].

**Table 2. Summary of qualitative findings.**

| Findings | Summary of qualitative review findings | Contributing qualitative studies | Overall CERQual assessment | Explanation of overall assessment |
|---|---|---|---|---|
| | Interplay between perceived risks and benefits of participation in women's decision-making | | | |
| 1 | **Women have a limited appetite for risk during pregnancy or lactation** Perception of risks influenced pregnant and lactating women's willingness to participate in trials, which varied based on their individual levels of risk tolerance, previous trial experiences, observations of others' experiences, stage of pregnancy or lactation, existing health conditions, and a sense of responsibility for their health and that of the fetus/infant. Women were more likely to decline participation if the experimental intervention was previously untested and were more confident to participate when convinced of no harm. | [39,40,47,48,57,58,60,63–65,69,72,74,83,84,87,89,91,92,96] | High confidence | No or very minor concerns on relevance, no or very minor concerns on coherence, no or very minor concerns on adequacy, minor concerns on methodological limitations (recruitment/data collection, analysis, link from data to findings, coherence on designs, ethics, reflexivity) |
| 2 | **Making trade-offs between risk and severity of the condition and risk-benefit ratio of intervention** Before participating, women weighed the risk of their medical condition and its impact, especially on the baby, against the risks of an intervention and its potential benefits. Women were less likely to participate if they felt healthy or perceived themselves at low risk of experiencing or being negatively affected by the condition, believed they had nothing to gain from participating, or felt concerned that the intervention risks were too high. | [39,48,57–60,63,64,69,72,74,87,91,96] | Moderate confidence | No or very minor concerns on methodological limitations, no or very minor concerns on coherence, minor concerns on adequacy (14 papers relatively thick data), moderate concerns on relevance (all are high-income countries) |
| 3 | **Benefits to health arising from participation** A key motivating factor for pregnant and lactating women to participate in trials was the expectation of personal health benefits, such as improved knowledge about how the condition affected them, protecting their fetus or infant from harm, and reducing mother-to-child disease transmission. When women saw the potential for these benefits, deciding not to participate was viewed as potentially putting the baby's life at risk. | [40,47,55,60,61,63,64,70,73,83,84,87,90–92,96] | High confidence | No or very minor concerns on relevance, no or very minor concerns on coherence, minor concerns on methodological limitations (recruitment/data collection, analysis, link from data to findings, coherence on designs, ethics, reflexivity), minor concerns on adequacy (16 papers with relatively thick data) |
| 4 | **Experiences and expectations of high-quality care motivate participation** Pregnant and lactating women were motivated to participate as a token of appreciation to health workers who provided good quality care. Additionally, women were more likely to participate when they perceived that it would result in higher quality clinical care or access to vaccines or therapeutic products that had previously been denied or were otherwise not accessible outside the context of a trial. | [39,48,49,60,63,70,72,83,84,86,87,92,96] | High confidence | No or very minor concerns on relevance, no or very minor concerns on coherence, minor concerns on adequacy (13 papers with relatively thin data), minor concerns on methodological limitations (recruitment/data collection, analysis, link from data to findings, coherence on designs, ethics, reflexivity) |

*(Continued)*

**Table 2.** (Continued)

| Findings | Summary of qualitative review findings | Contributing qualitative studies | Overall CERQual assessment | Explanation of overall assessment |
|---|---|---|---|---|
| 5 | **Knowledge of the rationale for study design features** The rationale behind certain trial design features such as randomisation, blinding, or inclusion of a placebo arm could be a source of confusion, concern, or reassurance for potential participants, impacting their decisions to participate. These features could be viewed as preferential treatment of one group over another, adding burden with little opportunity for personal benefit, a mechanism to reduce bias or conversely for researchers to avoid accountability for an adverse outcome. | [39,40,45,59,62,63,69,72,74,87,91,92] | Moderate confidence | No or very minor concerns on coherence, minor concerns on adequacy (12 papers contributed with relatively thick data), minor concerns on methodological limitations (recruitment/data collection, analysis, ethics, and reflexivity), moderate concerns on relevance (no papers conducted in middle-income countries and only 1 paper in a low-income country) |
| 6 | **Acceptability of the intervention is key to pregnant and lactating women's willingness to participate in a trial, and for research staff to recruit for a trial** Interventions that were most acceptable to women and research staff were those that simplified intervention delivery, were less onerous or painful than usual care, had negligible risk, were noninvasive, placed limited demands on time, did not involve invasive procedures, and where prior knowledge about the condition intersected with positive attitudes towards the therapeutic product. | [40,45,48,53,54,61,64,65,72,73,81,83,86,87,90–92,96] | High confidence | No or very minor concerns on methodological limitations, no or very minor concerns on coherence, minor concerns on adequacy (18 papers with relatively thick data), minor concerns on relevance (no papers conducted in upper middle-income countries) |
| 7 | **Fears around data sharing and use** Some women feared that trial participation, including provision of blood samples, could expose them to stigmatisation and judgement due to unwanted diagnoses and disclosure of disease status, data sharing regarding sensitive behaviours, and the threat of their data being used in ways that would compromise confidentiality and safety. | [65,85,86] | Low confidence | No or very minor concerns on methodological limitations, no or very minor concerns on coherence, moderate concerns on relevance (2 papers indirectly relevant to review aim, and no representation from middle-income countries), serious concerns on adequacy (3 papers with moderately thin data) |
| 8 | **Altruistic motivations** Pregnant women expressed willingness to participate in trials for the purpose of contributing to societal benefits of research, including the potential to improve health and healthcare for pregnant women in the future. Altruistic motivations could act as a stand-alone stimulus, secondary to or alongside beliefs around personal benefit, or conditional on no additional risk for participation. | [39,40,47,48,55–61,63,64,70,72–74,83,86,87,89,91,92] | Moderate confidence | No or very minor concerns on coherence, minor concerns on relevance (no studies conducted in low-income countries), minor concerns on adequacy (23 papers with relatively thin data), moderate concerns on methodological limitations (recruitment/data collection, analysis, link from data to findings, coherence on designs, ethics, and reflexivity) |
| 9 | **Financial incentives** Pregnant and lactating women had mixed attitudes to financial incentives for research participation. Some viewed financial incentives as acceptable, with higher remuneration as an appropriate strategy to encourage participation, whereas others viewed financial incentives as potentially coercive, especially in the context of poverty. Some women felt that financial reimbursements did not play a substantial role in women's decision-making. | [39,55,65,83,96] | Low confidence | No or very minor concerns on coherence, minor concerns on methodological limitations (recruitment/data collection, analysis, link from data to findings, coherence on designs, ethics, reflexivity), moderate concerns on relevance (no papers conducted in middle-income countries, and only 2 WHO regions and 3 countries represented), serious concerns on adequacy (5 papers with mostly thin data) |

(*Continued*)

**Table 2.** (Continued)

| Findings | Summary of qualitative review findings | Contributing qualitative studies | Overall CERQual assessment | Explanation of overall assessment |
|---|---|---|---|---|
| | **Engagement between women and the medical and research ecosystems** | | | |
| 10 | **Roles of trust and power in the medical and research ecosystem**<br>Pregnant and lactating women's willingness to participate in trials was driven by trust, confidence, and faith in medicine and research, and women relied on the opinions of the health workers that they consulted with regarding the efficacy and safety of the intervention. Simultaneously, power imbalances between women and health workers, coupled with women's therapeutic misconceptions, could lead to coercion in participation. This ethical dilemma was recognised by study investigators, ethics committee members, and women, especially in the context of the dual roles of clinician-researchers; however, power and credibility when combined with good rapport and clear communication generated trust to participate or comfort to decline. While rare, some women had larger concerns about the vested interests of pharmaceutical companies. | [39,40,42–45,47–49,56–61,65,69,70,72–74,81,82,86,87,89,91,92] | High confidence | No or very minor concerns on relevance, no or very minor concerns on coherence, minor concerns on methodological limitations (recruitment/data collection, analysis, link from data to findings, coherence on designs, ethics, reflexivity), minor concerns on adequacy (28 papers contributed to review findings with relatively thick data) |
| 11 | **The role of therapeutic hope and optimism**<br>Therapeutic hope and optimism played a critical role for health workers and research staff to administer trials, and for pregnant and lactating women to participate in trials. Prior knowledge about and experience with using the intervention, observation of potential beneficial effects, and trust in health workers shaped feelings of therapeutic hope and optimism. However, for some women, a lack of understanding of the differences between research and clinical care when combined with therapeutic hope led to therapeutic misconceptions and unmet expectations about the personal benefits arising from trial participation. | [42,45,47,53,65,70,74,81,82,87] | Moderate confidence | No or very minor concerns on coherence, no or very minor concerns on relevance, moderate concerns on adequacy (10 papers contributed with relatively thin data), moderate concerns on methodological limitations (recruitment/data collection, coherence on designs, ethics, reflexivity, analysis, link from data to findings) |
| | **Gender norms and decision-making autonomy** | | | |
| 12 | **Expectations of women's roles as mothers and caregivers**<br>Pregnant and lactating women's decisions to participate in clinical trials were often influenced by their strong sense of responsibility towards the health and care of their fetus or infant, themselves, and their families. This sense of responsibility was endorsed and reinforced by familial and societal expectations of what it means to be a good mother. | [60,61,64,91,96] | Low confidence | No or very minor concerns on methodological limitations, no or very minor concerns on coherence, serious concerns on adequacy (5 papers with relatively thin data), moderate concerns on relevance (no representation from low- and middle-income countries) |

*(Continued)*

**Table 2.** (Continued)

| Findings | Summary of qualitative review findings | Contributing qualitative studies | Overall CERQual assessment | Explanation of overall assessment |
|---|---|---|---|---|
| 13 | **Role of bodily autonomy in decision-making** Some women, health workers, ethics committee members, and regulators perceived that pregnant women might not be able to make decisions by themselves about trial participation due to fetal involvement, inability to make rational choices during pregnancy, hormones, the stressful context of hospitalisation, and financial inducements. However, research staff and some women believed in the right to bodily autonomy to make decisions by themselves despite having discussions with partners, family members, support persons, or health workers. Women viewed other people making decisions regarding their participation as a violation of this right, though some women declined participation due to pressure from family members. | [39,40,43,47,54,56,72,74,81,82,85,87,90,92] | Moderate confidence | No or very minor concerns on coherence, minor concerns on relevance (13 out 14 papers directly relevant to review aim), minor concerns on methodological limitations (recruitment/data collection, analysis, link from data to findings, coherence on designs, ethics, reflexivity), minor concerns on adequacy (14 papers with moderately thick data) |
| 14 | **Relationship dynamics, gender roles, and norms are key to women's attitudes to partner involvement and paternal consent** Pregnant women often discussed the benefits and risks of trial participation with their partners—especially in the context of fetal involvement—and their final decision may or may not have been influenced by their partners' own attitudes. In some settings, pregnant women's trial participation was contingent on partners' buy-in, and the formality justified in the context of gender norms and roles. These could be the partner being the household head, to allay men's suspicions about women's whereabouts and interactions, and to minimise any misunderstanding related to positive tests or disease status that might cast doubt on women's fidelity to their husbands. | [39,40,42,43,47,60,64,65,69,72,74,81,83,85,87,90,91] | Moderate confidence | No or very minor concerns on coherence, no or very minor concerns on relevance, minor concerns on methodological limitations (recruitment/data collection, analysis, link from data to findings, coherence on designs, ethics, reflexivity), moderate concerns on adequacy (17 papers contributed with relatively thin data) |
| | **Factors affecting clinical trial recruitment** | | | |
| 15 | **Developing trusting and reciprocal relationships with the community as part of the research process** Designing and embedding research within communities required engaging with community norms, beliefs, and practices. Some community members expressed how they viewed research negatively in the context of historical and ongoing oppressions that people experience due to colonisation, corruption, extractive practices, and civil and political conflict. Central to the acceptability and cultural safety of the research were investments in developing trusting relationships with community representatives and leaders. | [44,45,60,65,66,74,78,83,90,92] | Moderate confidence | No or very minor concerns on coherence, no or very minor concerns on relevance, minor concerns on methodological limitations (recruitment/data collection, analysis, coherence on designs, ethics, reflexivity), moderate concerns on adequacy (10 papers contributed with relatively thin data) |

*(Continued)*

**Table 2.** (Continued)

| Findings | Summary of qualitative review findings | Contributing qualitative studies | Overall CERQual assessment | Explanation of overall assessment |
|---|---|---|---|---|
| 16 | **Increasing visibility and awareness of the trial** Increasing visibility and awareness of the trial to potential participants, health workers, and community representatives influenced trial recruitment. Recommended strategies included paper and electronic promotional materials, regular physical presence of research staff in the areas where recruitment was taking place, and reminders to health workers about recruitment pathways and trial protocols through trainings. | [54,62,65,74,87] | Low confidence | No or very minor concerns on coherence, no or very minor concerns on methodological limitations, moderate concerns on relevance (contributing papers represented 3 regions where 4 are high-income countries and 1 low-income country), serious concerns on adequacy (5 papers contributed with relatively thick data) |
| 17 | **Inadequate resources** Inadequate physical infrastructure, time, finances, and insufficient quantity and quality of human resources were barriers for research staff to recruit women for clinical trials. For health workers specifically, heavy workloads made it challenging to incorporate trial recruitment into clinical workflows, and the added burden and sometimes insufficient compensation, contributed to poor morale. | [44,54,55,62,87,89] | Low confidence | No or very minor concerns on coherence, no or very minor concerns on relevance, moderate concerns on methodological limitations (recruitment/data collection, analysis, link from data to findings, coherence on designs, ethics, reflexivity), serious concerns on adequacy (6 papers contributed with relatively thin data) |
| 18 | **Engaging health workers in trials** Research staff perceived the importance of building reciprocal and collaborative relationships with health workers because some acted as gatekeepers. Some health workers, however, were reluctant to engage women in clinical trials due to a lack of knowledge about trial design and the research value, varying levels of acceptability of risk, perceived obligation to protect women, and a lack of trust in the research team. Health workers supported inclusion when trial protocols included close monitoring of risks and when there was clinical equipoise alongside therapeutic hope in the trial intervention. These factors were informed by their clinical knowledge, previous clinical experiences using the intervention, and observed outcomes in the current trial. | [47,53–55,60,62,64,65,87,89–91] | High confidence | No or very minor concerns on coherence, minor concerns on adequacy (12 papers contributed with thick data), minor concerns on relevance (contributing papers represented 10 countries with 8 high-income and 1 lower middle-income and 1 low-income country), minor concerns on methodological limitations (recruitment/data collection, analysis, link from data to findings, coherence on designs, ethics reflexivity) |
| 19 | **Research staff's emotional orientations towards clinical trials** Having a sense of trial ownership, supportive teamwork, a shared sense of team achievement and motivation to achieve recruitment targets could support successful trial recruitment. However, feeling pressured by the recruitment process, seeing it as a procedural activity and needing to implement complex study designs impacted research staffs' ability to recruit women, leading to frustration and lower enthusiasm. | [53,54,62] | Low confidence | No or very minor concerns on coherence, no or very minor concerns on methodological limitations, moderate concerns on adequacy (3 papers with thick data), serious concerns on relevance (papers represented 1 region where all countries are high-income countries) |

(*Continued*)

**Table 2.** (Continued)

| Findings | Summary of qualitative review findings | Contributing qualitative studies | Overall CERQual assessment | Explanation of overall assessment |
|---|---|---|---|---|
| 20 | **Women-centred approach encourages participation**<br>Women valued an individualised, humanised, and transparent approach to communication, and adequate time during trial recruitment to discuss details and concerns related to the trial. These helped ensure they had sufficient capacity and opportunity to make informed decisions. Similarly, research staff found that approaching potential participants at the "right time" and in an appropriate manner by considering their physical and mental state, providing adequate information and engaging in discussions increased recruitment success. | [39,40,54,56,62,66,69,70,72,74,86,87,92] | Moderate confidence | No or very minor concerns on coherence, minor concerns on methodological limitations (recruitment/data collection, analysis, link from data to findings, coherence on designs, ethics, reflexivity), minor concerns on adequacy (13 papers contributed with mostly thick data), moderate concerns on relevance (no representation from low-income countries) |
| 21 | **Recruitment for intrapartum research**<br>Pain, intensity, and duration of labour motivated pregnant women to participate in intrapartum clinical trials. However, women, their partners, and research staff recognised the challenges in ensure women make informed decisions during this sensitive time, as decisions had to be made quickly, and partners were reluctant to make decisions on women's behalf, even during emergencies, due to fears of negative outcomes. To optimise women making informed decisions, research staff provided information clearly and succinctly during the intrapartum period and tried to offer adequate time for decision-making. Most women recommended having trial information provided in the antenatal period and revisiting trial details, including having a de-briefing about one's own experience, prior to discharge. | [43,49,56,59,61,62,81,82,86,91] | Moderate confidence | No or very minor concerns on coherence, minor concerns on adequacy (10 papers contributed with relatively thick data), minor concerns on methodological limitations (recruitment/data collection, analysis, link from data to findings, coherence on designs, ethics, reflexivity), moderate concerns on relevance (9 out of 10 papers indirectly relevant, all papers are high-income countries) |
| | **Upstream factors affecting the research ecosystem** | | | |
| 22 | **Factors affecting motivation of study investigators**<br> The underlying factors that motivated many study investigators to conduct research with pregnant women were ethical responsibility, passion towards equity, and dedication to improving women's health status and care, and filling scientific gaps. Additionally, lived experience of being pregnant, having mentors in this area in early careers, and previous research experiences with pregnant women contributed to study investigators' motivations. However, concerns about risks of teratogenicity demotivated some investigators. | [42,43,66,78,89,91] | Moderate confidence | No or very minor concerns on coherence, no or very minor concerns on relevance, minor concerns on adequacy (6 papers contributed with relatively thick data), moderate concerns on methodological limitations (recruitment/data collection, analysis, link from data to findings, coherence on designs, ethics, reflexivity) |

*(Continued)*

**Table 2.** (Continued)

| Findings | Summary of qualitative review findings | Contributing qualitative studies | Overall CERQual assessment | Explanation of overall assessment |
|---|---|---|---|---|
| 23 | **Challenges in gaining ethical approvals for trials with pregnant women** While some regulators, ethics committee members, and study investigators strongly support inclusion of pregnant women in clinical trials, most stakeholders start from a presumption of minimal risk to the fetus. This results in women's exclusion, especially in the context of poor public stewardship, ambiguous guidelines, insufficient data on intervention safety, complexities and subjectivities in risk assessment, poor agreement on appropriate trial design, time consuming ethical processes, and concerns about reputation. | [42,43,66,78,82,89–91] | Moderate confidence | No or very minor concerns on coherence, no or very minor concerns on relevance, minor concerns on adequacy (8 papers contributed with relatively thick data), minor concerns on methodological limitations (recruitment/data collection, analysis, link from data to findings, coherence on designs, ethics, reflexivity) |
| 24 | **Role of funders** Limited interest of public and private funders and pharmaceutical companies to financially invest in trials due to the ethical complexities, potential for adverse events, liability, and possibility of political fallout was a barrier to conduct trials with pregnant and lactating women. When funding was available, funders' requests might facilitate the inclusion of pregnant women or create ethical challenges in conducting trials. | [54,62,66,78] | Low confidence | No or very minor concerns on coherence, no or very minor concerns on relevance, moderate concerns on methodological limitations (recruitment/data collection, analysis, link from data to findings, ethics, reflexivity), serious concerns on adequacy (4 papers contributed with relatively thin data) |

1. interplay between perceived risks and benefits of participation in women's decision-making (9 review findings);

2. engagement between women and the medical and research ecosystems (2 review findings);

3. gender norms and decision-making autonomy (3 review findings);

4. factors affecting clinical trial recruitment (7 review findings); and

5. upstream factors in the research ecosystem (3 review findings).

We graded 6 review findings as high confidence, 11 as moderate confidence, and 7 as low confidence. An explanation for each GRADE-CERQual assessment is presented in the evidence profile (S4 Appendix).

## Interplay between perceived risks and benefits of participation in women's decision-making

Findings 1 to 9 are categorised under this theme with 48 studies exploring women's perspectives on clinical trial participation and factors influencing their decision-making. These factors include balancing risks and benefits, experiences and expectations of high quality care, understanding of study design features, acceptability and stigma associated with the intervention, altruistic motivations and financial incentives.

*Finding 1*: *Women have a limited appetite and higher perception of risk during pregnancy or lactation.* **Perception of risks influenced pregnant and lactating women's willingness to participate in trials, which varied based on their individual levels of risk tolerance, previous trial experiences, observations of others' experiences, stage of pregnancy or lactation, existing health conditions, and a sense of responsibility for their health and that of the fetus/infant. Women were more likely to decline participation if the experimental intervention was previously untested and were more confident to participate when convinced of no harm** (high confidence) [39,40,47,48,57,58,60,63–65,69,72,74,83,84,87,89,91,92,96].

The most salient factors affecting perceptions of risk were concerns of potential harm to the fetus or baby, including in the longer term, and fears of side-effects [39,48,57,58,60,63,69,72,74,83,84,87,89,91,92,96]. The uncertainty of these negative outcomes contributed to women's reluctance to take medications [48,64,69,72] or participate in experimental interventions, with some likening the experience to being treated as "guinea pigs" [39,56,58,69,90]. Women willing to consider participation wanted proof of safety from previous research evidence [57,58,84], online resources [96], discussions with research staff and health workers [96], and knowing the experiences of others who had taken the intervention [47,96].

Quantitative evidence supported the qualitative findings that women were apprehensive about taking an experimental product during pregnancy or lactation [79] primarily due to concerns of fetal or infant harm [38,51,67,71,75,83,94,95], side-effects [77,80], and the possibility of unknown longer-term negative sequelae [67,75,77]. Prior knowledge of the health condition [68], information about drug safety in pregnant and nonpregnant populations [51], and information that large numbers of pregnant women had already enrolled in the trial [67] were factors that increased willingness to participate.

*Finding 2*: *Making trade-offs between risk and severity of the condition and risk-benefit ratio of intervention.* **Before participating, women weighed the risk of their medical condition and its impact, especially on the baby, against the risks of an intervention and its potential benefits. Women were less likely to participate if they felt healthy or perceived themselves at low risk of experiencing or being negatively affected by the condition, believed they had nothing to gain from participating, or felt concerned that the intervention risks were too high** (moderate confidence) [39,48,57–60,63,64,69,72,74,87,91,96].

Women were more willing to participate when they had concerns about their risk factors [70], had previously experienced the condition [48,70], or personally knew someone who had [48], were anxious about the baby suffering health problems [57–60], or perceived the intervention to be helpful based on past use [87], or the only course of action to avoid (further) ill-health [57–59,63,91]. For some women with preconceived notions that research entailed significant risks, their perceptions did not change in the presence of information, including about intervention safety [48].

Quantitative evidence supported the qualitative findings that, when coupled with risks that were considered minimal or manageable [83], women with greater knowledge about [83] or direct exposure to the condition [94] were more likely to participate in a vaccine or therapeutic trial. However, prior exposure to the medical condition did not consistently lead to higher participation in trials [51].

*Finding 3*: *Benefits to health arising from participation.* **A key motivating factor for pregnant and lactating women to participate in trials was the expectation of personal health benefits, such as improved knowledge about how the condition affected them, protecting their fetus or infant from harm, and reducing mother-to-child disease transmission. When**

**women saw the potential for these benefits, deciding not to participate was viewed as potentially putting the baby's life at risk** (high confidence) [40,47,55,60,61,63,64,70,73,83,84,87,90–92,96].

Quantitative evidence supported this finding that women were more willing to participate in a trial when they were convinced about the potential short and longer-term benefits of the intervention for the health of the fetus [38,51,75,77,80], and their own health [38,41,51,75,80,95] and education [41,95].

*Finding 4*: *Experiences and expectations of high-quality care motivate participation.* **Pregnant and lactating women were motivated to participate as a token of appreciation to health workers who provided good quality care. Additionally, women were more likely to participate when they perceived that it would result in higher quality clinical care or access to vaccines or therapeutic products that had previously been denied or were otherwise not accessible outside the context of a trial** (high confidence) [39,48,49,60,63,70,72,83,84,86,87,92,96].

In addition to free medications and vaccines, women's perceptions of higher quality care were linked to greater frequency of diagnostic and monitoring tests [72,83,84,92], detailed information regarding care provided [63], and closer and continuous clinical observation [49,63,70,92]. Occasionally, women perceived care associated with a trial as lower quality due to the "experimental" nature of the intervention [39].

Quantitative evidence supported the qualitative finding that women expected trial participation to engender more and better quality care through enhanced monitoring [38,41,67,68,80], more tests [67], better therapeutic treatment [38,49], and the general feeling of being provided a high standard of medical care [51,75,80].

*Finding 5*: *Knowledge of the rationale for study design features.* **The rationale behind certain trial design features such as randomisation, blinding or inclusion of a placebo arm could be a source of confusion, concern, or reassurance for potential participants, impacting their decisions to participate. These features could be viewed as preferential treatment of one group over another, adding burden with little opportunity for personal benefit, a mechanism to reduce bias or conversely for researchers to avoid accountability for an adverse outcome** (moderate confidence) [39,40,45,59,62,63,69,72,74,87,91,92].

Quantitative evidence extended understanding of women's views about participation in placebo-controlled trials. Some women expressed reluctance to participate due to the possibility of being assigned to the control or placebo group [67,77,79,83]. However, others expressed that the uncertainty of assignment would not affect their decision, and for a minority, the possibility of assignment to the control condition motivated their participation as it could minimise risk but still provide ancillary benefits [67]. Women were keen to be unblinded regarding the arm to which they were assigned, once the trial was complete [80].

*Finding 6*: *Acceptability of the intervention is key to pregnant and lactating women's willingness to participate in a trial and for research staff to recruit for a trial.* **Interventions that were most acceptable to women and research staff were those that simplified intervention delivery, were less onerous or painful than usual care, had negligible risk, were noninvasive, placed limited demands on time, did not involve invasive procedures, and where prior knowledge about the condition intersected with positive attitudes towards the therapeutic product** (high confidence) [40,45,48,53,54,61,64,65,72,73,81,83,86,87,90–92,96].

For health workers involved in recruitment and trial operations, acceptability of the intervention was closely linked to their perceptions of the safety of the experimental therapy,

derived from previous positive experiences administering the drug in a different clinical setting [53].

Quantitative evidence supported this qualitative finding that some women might be more willing to participate in a trial when they were less likely to be inconvenienced by or experience discomfort from trial procedures, additional and lengthy study visits [38,41,80]. Decliners cited blood tests, additional scans, and availability of suitable noninvasive alternatives as reasons for nonparticipation [51,83]. In the case of vaccine trials, quantitative data extended this qualitative finding by suggesting that women indicated greater acceptability of inactivated virus vaccines compared to live-attenuated virus vaccines [51].

*Finding 7*: *Fears around data sharing and use*. **Some women feared that trial participation, including provision of blood samples, could expose them to stigmatisation and judgement due to unwanted diagnoses and disclosure of disease status, data sharing regarding sensitive behaviours, and the threat of their data being used in ways that would compromise confidentiality and safety** (low confidence) [65,85,86]. In the context of HIV trials, some women discussed concerns that an HIV diagnosis would lead to abandonment by their husbands [85].

No quantitative evidence was identified in this domain.

*Finding 8*: *Altruistic motivations*. **Pregnant women expressed willingness to participate in trials for the purpose of contributing to societal benefits of research, including the potential to improve health and healthcare for pregnant women in the future. Altruistic motivations could act as a stand-alone stimulus, secondary to or alongside beliefs around personal benefit, or conditional on no additional risk for participation** (moderate confidence) [39,40,47,48,55–61,63,64,70,72–74,83,86,87,89,91,92].

In addition to helping other women, altruistic sentiments were linked to perceptions that the research effort was worthy [48,59,61], well-intentioned [61], filled an important scientific gap [58,70,72], and addressed a pressing need [48,63,73,91].

Quantitative evidence supported the qualitative finding that altruistic motivations influenced willingness to participate in trials, alongside personal benefits [38,41,49,51,67,77,80,95]. Women expressed having a sense of fulfilment that participation would have a positive impact on women's health in the future.

*Finding 9*: *Financial incentives*. **Pregnant and lactating women had mixed attitudes to financial incentives for research participation. Some viewed financial incentives as acceptable, with higher remuneration as an appropriate strategy to encourage participation, whereas others viewed financial incentives as potentially coercive, especially in the context of poverty. Some women felt that financial reimbursements did not play a substantial role in women's decision-making** (low confidence) [39,55,65,83,96].

Negative views on renumeration arose from concerns that financial incentives would entice women to enrol multiple times [65], or make it challenging for them to withdraw from the study [39].

Quantitative evidence extended this qualitative finding by suggesting that attitudes to financial compensation differed based on levels of education attainment [97]. In one study, less than 1 in 10 women discussed that financial incentives would increase their likelihood of participation in medication or vaccine-based research [75], whereas in another, 4 in 10 women agreed that they volunteered to participate due to financial compensation [41].

## Engagement between women and the medical and research ecosystems

Findings 10 and 11 are categorised under this theme, with 34 contributing studies examining factors operating at the intersection of women and the medical and research ecosystems. The

factors include women's reliance on health workers' clinical opinions to assist decision-making, and the role of therapeutic hope and optimism in women's decisions to participate and health worker and research staffs' motivations to administer trials.

*Finding 10*: *Roles of trust and power in the medical and research ecosystem*. **Pregnant and lactating women's willingness to participate in trials was driven by trust, confidence, and faith in medicine and research, and women relied on the opinions of the health workers that they consulted with regarding the efficacy and safety of the intervention. Simultaneously, power imbalances between women and health workers, coupled with women's therapeutic misconceptions, could lead to coercion in participation. This ethical dilemma was recognised by study investigators, ethics committee members, and women, especially in the context of the dual roles of clinician-researchers; however, power and credibility when combined with good rapport and clear communication generated trust to participate or comfort to decline. While rare, some women had larger concerns about the vested interests of pharmaceutical companies** (high confidence) [39,40,42–45,47–49,56–61,65,69,70,72–74,81,82,86,87,89,91,92].

Quantitative data supported the qualitative finding that trust (or lack thereof) in health workers, research teams, and pharmaceutical companies affected participation [38,51,75,95]. Some women felt pressured to participate by health workers and were disappointed by the lack of an individualised approach to recruitment [80]. Among decliners of a vaccine trial, some noted that recommendations from a health worker could motivate a change of mind [51].

*Finding 11*: *The role of therapeutic hope and optimism*. **Therapeutic hope and optimism played a critical role for health workers and research staff to administer trials, and for pregnant and lactating women to participate in trials. Prior knowledge about and experience with using the intervention, observation of potential beneficial effects, and trust in health workers shaped feelings of therapeutic hope and optimism. However, for some women, a lack of understanding of the differences between research and clinical care when combined with therapeutic hope led to therapeutic misconceptions and unmet expectations about the personal benefits arising from trial participation** (moderate confidence) [42,45,47,53,65,70,74,81,82,87].

Health workers expressed the importance of women and themselves comprehending the differences between research and clinical care to minimise participation arising from therapeutic misconceptions [47].

No quantitative evidence was identified in this domain.

## Gender norms and decision-making autonomy

Findings 12 to 14 are categorised under this theme with 24 contributing studies discussing women's roles as mothers and caregivers, mixed perceptions of women's autonomous decision-making, and intimate male partner involvement in decision-making.

*Finding 12*: *Expectations of women's roles as mothers and caregivers*. **Pregnant and lactating women's decisions to participate in clinical trials were often influenced by their strong sense of responsibility towards the health and care of their fetus or infant, themselves, and their families. This sense of responsibility was endorsed and reinforced by familial and societal expectations of what it means to be a good mother** (low confidence) [60,61,64,91,96].

For some women, this responsibility to protect their baby translated to not engaging in any actions that might risk jeopardising the baby's health [91].

No quantitative evidence was identified in this domain.

*Finding 13*: *Role of bodily autonomy in decision-making.* **Some women, health workers, ethics committee members, and regulators perceived that pregnant women might not be able to make decisions by themselves about trial participation due to fetal involvement, inability to make rational choices during pregnancy, hormones, the stressful context of hospitalisation and financial inducements. However, research staff and some women believed in the right to bodily autonomy to make decisions by themselves despite having discussions with partners, family members, support persons, or health workers. Women viewed other people making decisions regarding their participation as a violation of this right, though some women declined participation due to pressure from family members** (moderate confidence) [39,40,43,47,54,56,72,74,81,82,85,87,90,92].

Women also believed that research could be an avenue through which women demanded their rights in the healthcare [65].

Quantitative evidence supported qualitative findings that women believed in their capability to make decisions regarding trial participation, with some doing so autonomously and others receiving support from family members [38,83].

*Finding 14*: *Relationship dynamics, gender roles, and norms are key to women's attitudes to partner involvement and paternal consent.* **Pregnant women often discussed the benefits and risks of trial participation with their partners—especially in the context of fetal involvement—and their final decision may or may not have been influenced by their partners' own attitudes. In some settings, pregnant women's trial participation was contingent on partners' buy-in, and the formality justified in the context of gender norms and roles. These could be the partner being the household head, to allay men's suspicions about women's whereabouts and interactions, and to minimise any misunderstanding related to positive tests or disease status that might cast doubt on women's fidelity to their husbands** (moderate confidence) [39,40,42,43,47,60,64,65,69,72,74,81,83,85,87,90,91].

Partner involvement was not preferred when that partner was abusive or uninvolved, or when a woman was unmarried, or the pregnancy had occurred in the context of rape [85]. Furthermore, imposing a paternal consent rule in these circumstances was a serious barrier to participation [85]. When research participation violated gender roles and norms, it sometimes resulted in partner violence, marital breakdown, or rejection of the baby [85].

No quantitative evidence was identified in this domain.

## Factors affecting clinical trial recruitment

Findings 15 through 21 are categorised under this theme with 41 contributing studies exploring the importance of cultural acceptability and safety of intervention procedures, development of reciprocal relationships between research staff and health workers, the importance of resource availability, trial visibility and emotional orientations, and woman-centred approach to recruitment.

*Finding 15*: *Developing trusting and reciprocal relationships with the community as part of the research process.* **Designing and embedding research within communities required engaging with community norms, beliefs, and practices. Some community members expressed how they viewed research negatively in the context of historical and ongoing oppressions that people experience due to colonisation, corruption, extractive practices, and civil and political conflict. Central to the acceptability and cultural safety of the research were investments in developing trusting relationships with community representatives and leaders** (moderate confidence) [44,45,60,65,66,74,78,83,90,92].

This was achieved through dialogue and engagement starting at research conceptualisation, collaborating with community representatives and previous research participants to develop communication and mobilisation strategies, providing accurate information about study procedures, and ensuring alignment of these procedures with community norms, beliefs, and practices.

No quantitative evidence was identified in this domain.

*Finding 16*: *Increasing visibility and awareness of the trial.* **Increasing visibility and awareness of the trial to potential participants, health workers, and community representatives influenced trial recruitment. Recommended strategies included paper and electronic promotional materials, regular physical presence of research staff in the areas where recruitment was taking place, and reminders to health workers about recruitment pathways and trial protocols through trainings** (low confidence) [54,62,65,74,87].

Quantitative evidence extended the qualitative finding that women preferred to have information about trials through their health workers [67].

*Finding 17*: *Inadequate resources.* **Inadequate physical infrastructure, time, finances, and insufficient quantity and quality of human resources were barriers for research staff to recruit women for clinical trials. For health workers specifically, heavy workloads made it challenging to incorporate trial recruitment into clinical workflows, and the added burden and sometimes insufficient compensation, contributed to poor morale** (low confidence) [44,54,55,62,87,89].

In terms of competency of human resources, research staff shared that their recruiting capability was built through practice and working alongside more experienced colleagues [54]. A key limiting factor in the recruitment of women from non-English speaking backgrounds was the unavailability of interpreters [87].

Quantitative evidence similarly reported that lack of infrastructure and limited time due to heavy workloads for health workers were barriers to including pregnant women in trials [50,67,88].

*Finding 18*: *Engaging health workers in trials.* **Research staff perceived the importance of building reciprocal and collaborative relationships with health workers because some acted as gatekeepers. Some health workers, however, were reluctant to engage women in clinical trials due to a lack of knowledge about trial design and the research value, varying levels of acceptability of risk, perceived obligation to protect women, and a lack of trust in the research team. Health workers supported inclusion when trial protocols included close monitoring of risks and when there was clinical equipoise alongside therapeutic hope in the trial intervention. These factors were informed by their clinical knowledge, previous clinical experiences using the intervention, and observed outcomes in the current trial** (high confidence) [47,53–55,60,62,64,65,87,89–91].

Quantitative evidence supported qualitative findings that knowledge of the relevance, feasibility, and ethical obligations to include pregnant and lactating women in trials, perceptions that pregnant women were a vulnerable population, lack of interest in trials, and preferences for noninvasive treatment were factors influencing whether health workers encouraged pregnant women's clinical trial participation [50,52,67,88,94,95].

*Finding 19*: *Research staff's emotional orientations towards clinical trials.* **Having a sense of trial ownership, supportive teamwork, a shared sense of team achievement and motivation to achieve recruitment targets could support successful trial recruitment. However, feeling**

**pressured by the recruitment process, seeing it as a procedural activity and needing to implement complex study designs impacted research staffs' ability to recruit women, leading to frustration and lower enthusiasm** (low confidence) [53,54,62].

No quantitative evidence was identified in this domain.

*Finding 20*: *Women-centred approach encourages participation.* **Women valued an individualised, humanised, and transparent approach to communication, and adequate time during trial recruitment to discuss details and concerns related to the trial. These helped ensure they had sufficient capacity and opportunity to make informed decisions. Similarly, research staff found that approaching potential participants at the "right time" and in an appropriate manner by considering their physical and mental state, providing adequate information and engaging in discussions increased recruitment success** (moderate confidence) [39,40,54,56,62,66,69,70,72,74,86,87,92].

To support an individualised recruitment approach, research staff reviewed obstetric information from women's charts [54,86] and had discussions with health workers [86] to tailor the recruitment information to women's personal situations. They also discussed using intuition to determine when and whom to approach for trial participation [54], considering the extent to which women looked sick or unwell at the time of recruitment [86].

Quantitative data supported this qualitative finding of women noting the significance of having detailed and well-explained trial information, including about risks and benefits, and adequate time to make decisions regarding participation [80,95]. Some women expressed disappointment when they felt they had been ill-informed about study procedures by research staff [80].

*Finding 21*: *Recruitment for intrapartum research.* **Pain, intensity, and duration of labour motivated pregnant women to participate in intrapartum clinical trials. However, women, their partners, and research staff recognised the challenges in ensure women make informed decisions during this sensitive time, as decisions had to be made quickly, and partners were reluctant to make decisions on women's behalf, even during emergencies, due to fears of negative outcomes. To optimise women making informed decisions, research staff provided information clearly and succinctly during the intrapartum period and tried to offer adequate time for decision-making. Most women recommended having trial information provided in the antenatal period, and revisiting trial details, including having a de-briefing about one's own experience, prior to discharge** (moderate confidence) [43,49,56,59,61,62,81,82,86,91].

Quantitative data extended this qualitative finding with most ethics committee members considering consent in-labour as ethical. Factors that ethics committee members considered when approving labour trials, included the level of risk involved and women's ability to provide informed consent [76]. Most ethics committee members also supported the involvement of partners in the consent process [76]. Aligned with the qualitative data, women expressed a preference to be approached for a labour trial earlier to have adequate time for discussion and an informed decision [79,80].

## Upstream factors affecting the research ecosystem

Findings 22 to 24 are categorised under this theme with 13 studies discussing factors operating at the level of study investigators, ethics committees, and funders. The factors include study investigators' personal and professional motivations to pursue research with pregnant women, complexities in obtaining ethical approval, and limited interest of funders to support clinical

trials with pregnant and lactating women.

*Finding 22*: *Factors affecting motivation of study investigators.* **The underlying factors that motivated many study investigators to conduct research with pregnant women were ethical responsibility, passion towards equity, and dedication to improving women's health status and care, and filling scientific gaps. Additionally, lived experience of being pregnant, having mentors in this area in early careers, and previous research experiences with pregnant women contributed to study investigators' motivations. However, concerns about risks of teratogenicity demotivated some investigators** (moderate confidence) [42,43,66,78,89,91].

No quantitative evidence was identified in this domain.

*Finding 23*: *Challenges in gaining ethical approvals for trials with pregnant women.* **While some regulators, ethics committee members, and study investigators strongly support inclusion of pregnant women in clinical trials, most stakeholders start from a presumption of minimal risk to the fetus. This results in women's exclusion, especially in the context of poor public stewardship, ambiguous guidelines, insufficient data on intervention safety, complexities and subjectivities in risk assessment, poor agreement on appropriate trial design, time-consuming ethical processes, and concerns about reputation** (moderate confidence) [42,43,66,78,82,89–91].

Study investigators and ethics committee members reported that these challenges could be overcome through shared institutional commitment to pregnant women's inclusion, close collaboration between investigators and ethics committee members from protocol inception, mutual understanding about each other roles, responsibilities, and intentions, development and implementation of practical guidance for consistency in regulatory interpretation and risk assessment, safety monitoring during implementation, and safeguards for injury compensation [42,66,78,89,91].

Quantitative evidence supported qualitative findings that obtaining regulatory approval for clinical trials that include pregnant women was challenging [88] due to ethics committees' preference for observational studies over trials [93], and varied opinions on the inclusion of pregnant women and what constituted minimal risk [76,93]. Most ethics committee members were also aware that they did not have adequate policy or guidance to inform their decisions to ensure equitable subject selection [76,93].

*Finding 24*: *Role of funders.* **Limited interest of public and private funders and pharmaceutical companies to financially invest in trials due to the ethical complexities, potential for adverse events, liability, and possibility of political fallout was a barrier to conduct trials with pregnant and lactating women. When funding was available, funders' requests might facilitate the inclusion of pregnant women or create ethical challenges in conducting trials** (low confidence) [54,62,66,78].

No quantitative evidence was identified in this domain.

## Mapping review findings to TDF, COM-B, and potential implementation strategies

Table 3 presents the mapping of review findings to the applicable TDF [24] and COM-B model domains [25], and the BCW intervention types to inform proposed strategies that address these factors. The strategies that we have identified are designed to provide a theoretically informed guide to the types of actions that can be taken to address barriers at various

Table 3. Mapping qualitative findings to the TDF and COM-B frameworks and BCW.

| Findings | Review findings | COM-B and TDF mapping | Actor | Potential intervention type based on BCW | Examples of how implementation strategies could be operationalised or actual implementation strategies used |
|---|---|---|---|---|---|
| 1 | **Women have a limited appetite for risk during pregnancy or lactation** | **Psychological Capability** (Memory, Attention and Decision processes); **Reflective Motivation** (Beliefs about Consequences) | Pregnant and lactating women, health workers, and research staff | Education, Persuasion, Enablement, Training | 1) For health workers, provide access to credible resources of empirical evidence or proof related to risks, benefits, side-effects, how past and ongoing medical complications might be affected by trial intervention, and probability and implications of possible interactions between current medications and the experimental product being tested through the trial that can support addressing fear, uncertainty, and provision of clinical advice around medication use during pregnancy. 2) Provide training to researchers on how to humanise engagement between the study team and participants through all trial stages. Training could include how to apply an individualised approach to recruitment that considers women's medical conditions, their individual and family situations, and preferences for engagement. Additionally, develop mechanisms for obtaining and integrating participant feedback on interpersonal engagement through all trial stages. 3) For women, see review finding 1.2. |
| 2 | **Making trade-offs between risk and severity of the condition and risk-benefit ratio of intervention** | **Psychological Capability** (Memory, Attention, and Decision processes); **Reflective Motivation** (Beliefs about Consequences, Optimism) | Pregnant and lactating women and research staff | Education, Training, Enablement | 1) For pregnant and lactating women, share information transparently about safety, risks, benefits, and side effects of the trial intervention, and encourage personalised discussions about how these aspects relate to women's own perceptions about participation and their individual situations to increase their knowledge and understanding. Inform women about the number of participants that have already enrolled in the trial and if the intervention has been tested in other settings. Engage prior trial participants to discuss personal experiences of trial participation. 2) For research staff, see review finding 1.1. |

(*Continued*)

**Table 3.** (Continued)

| Findings | Review findings | COM-B and TDF mapping | Actor | Potential intervention type based on BCW | Examples of how implementation strategies could be operationalised or actual implementation strategies used |
|---|---|---|---|---|---|
| 3 | **Benefits to health arising from participation** | **Reflective Motivation** (Beliefs about Consequences, Optimism) | Pregnant and lactating women | Education | For pregnant and lactating women, consider how sharing information about the trial can be tailored to their personal circumstances, for example, with the assistance of a decision-analysis tool. See also review finding 1.2. |
| 4 | **Experiences and expectations of high-quality care motivate participation** | **Reflective Motivation** (Intentions, Beliefs about Consequences); **Automatic Motivation** (Emotion) | Pregnant and lactating women | Education, Enablement | Consider how trusted resources, such as the woman's care provider, could help to communicate with women about benefits and risks of trial participation. |
| 5 | **Knowledge of the rationale for study design features** | **Psychological Capability** (Knowledge); **Reflective Motivation** (Beliefs about Consequences); **Automatic Motivation** (Emotion) | Pregnant and lactating women and research staff | Education, Training, Enablement | Provide clear explanations for the trial design features in plain language in the recruitment materials, informed consent form, and through discussion with women. Ensure research staff can clearly explain the rationale for different trial design features using plain language. |
| 6 | **Acceptability of the intervention is key to pregnant and lactating women's willingness to participate in a trial, and for research staff to recruit for a trial** | **Psychological Capability** (Knowledge); **Reflective Motivation** (Beliefs about Consequences, Optimism) | Pregnant and lactating women and research staff | Education, Training, Enablement | Conduct formative research before the start of the trial to assess the acceptability of different intervention components, and what components may need adjustments. Consider how trial procedures and intervention components can be simplified or streamlined to improve acceptability. |
| 7 | **Fears around data sharing and use** | **Psychological Capability** (Knowledge); **Reflective Motivation** (Beliefs about Consequences); **Physical Opportunity** (Environmental Context and Resources); **Social Opportunity** (Social Influences) | Pregnant and lactating women and community leaders/members | Education, Training, Enablement, Environmental restructuring | Ensure that data management plans include efforts to preserve data confidentiality. Train all health workers and research staff on the importance of maintaining confidentiality. Consider how research materials (e.g., forms, packaging) and the research environment (e.g., signposting, area of health facility) can be structured to prevent disclosure of stigmatising health conditions. Community engagement and outreach can also help to de-stigmatise health conditions. |
| 8 | **Altruistic motivations** | **Reflective Motivation** (Intentions, Beliefs about Consequences) | Pregnant and lactating women | Education | Provide information about the potential societal benefits of the research to women during recruitment and throughout the trial. |

(*Continued*)

**Table 3.** (Continued)

| Findings | Review findings | COM-B and TDF mapping | Actor | Potential intervention type based on BCW | Examples of how implementation strategies could be operationalised or actual implementation strategies used |
|---|---|---|---|---|---|
| 9 | **Financial incentives** | **Automatic Motivation** (Reinforcement) | Pregnant and lactating women | Enablement | Conduct formative research before the start of the trial to consider different perspectives on appropriate and ethical remuneration for trial participation. Consider how nonfinancial remuneration may also be acceptable (e.g., providing childcare during trial visits, providing transportation vouchers). |
| 10 | **Roles of trust and power in the medical and research ecosystem** | **Psychological Capability** (Knowledge, Memory, Attention and Decision process); **Reflective Motivation** (Beliefs about Consequences, Optimism) | Pregnant and lactating women and health workers | Education, Training, Enablement | Provide training to health workers involved in the trial that elaborates on the trial rationale, potential benefits, and where the trial fits into existing evidence. Ensure that women clearly understand the context of a trial and that participation may not yield therapeutic benefit. Where benefit is demonstrated in a trial, consider how participants randomised to a control group may receive the intervention at a later stage, or how scale-up could happen in control sites. |
| 11 | **The role of therapeutic hope and optimism** | **Reflective Motivation** (Social/ Professional Role and Identity); **Social Opportunity** (Social Influences) | Pregnant and lactating women, family members and community members | Education, Enablement, Persuasion | Provide training to health workers involved in the trial that elaborates on the trial rationale, potential benefits, and where the trial fits into existing evidence. Ensure that women clearly understand the context of a trial and that participation may not yield therapeutic benefit. Where benefit is demonstrated in a trial, consider how participants randomised to a control group may receive the intervention at a later stage, or how scale-up could happen in control sites. |
| 12 | **Expectations of women's roles as mothers and caregivers** | **Psychological Capability** (Knowledge, Memory, Attention, and Decision process); **Reflective Motivation** (Beliefs about Consequences, Optimism) | Pregnant and lactating women, health workers, and research staff | Education, Training, and Enablement | Ensure that information about the trial considers gendered aspects of motherhood and balances gendered beliefs and responsibilities with clear information about trial participation. Provide educational materials about pregnancy and the trial to women to share with their partners/spouses and family members. |

(*Continued*)

**Table 3.** (*Continued*)

| Findings | Review findings | COM-B and TDF mapping | Actor | Potential intervention type based on BCW | Examples of how implementation strategies could be operationalised or actual implementation strategies used |
|---|---|---|---|---|---|
| 13 | **Role of bodily autonomy in decision-making** | **Reflective Motivation** (Beliefs about Capabilities); **Social Opportunity** (Social Influences) | Pregnant and lactating women, family members, health workers, research staff, ethics committee members, and regulators | Education, Enablement | Develop training materials for ethics committee members and regulators about the ethical conduct of interventional research with women during pregnancy. Consider societal norms in the trial context to ensure that women are provided with the opportunity to accept or decline participation without undue influence from other people. |
| 14 | **Relationship dynamics, gender roles, and norms are key to women's attitudes to partner involvement and paternal consent** | **Reflective Motivation** (Beliefs about Consequences); **Social Opportunity** (Social Influences) | Pregnant and lactating women and partners | Education, Enablement | Prior to trial recruitment, ensure that research staff have a clear picture of social and gender norms around women's healthcare-seeking behaviours during pregnancy and birth. Engage with patient advocates and women's groups to ensure that trial procedures and recruitment processes are respectful of and responsive to social and gender norms. Develop decision aids and educational materials that can be used to foster discussion about participation between a woman and her spouse/partner and family, as appropriate. |
| 15 | **Developing trusting and reciprocal relationships with the community as part of the research process** | **Automatic Motivation** (Emotion); **Physical Opportunity** (Environmental Context and Resources); **Social Opportunity** (Social Influences) | Pregnant and lactating women, community members, health workers, and research staff | Training, Enablement, Environmental restructuring | Engage with patient advocate and women's groups throughout the trial to improve design, recruitment, implementation, and dissemination efforts. Patient advocate and women's groups can help to improve the person-centredness and inclusivity of research and develop strategies to address stigma or mistrust associated with clinical trials. Depending on context, engaging with community health workers or community leaders may help to align trial activities with community norms and beliefs. |
| 16 | **Increasing visibility and awareness of the trial** | **Psychological Capability** (Knowledge); **Physical Opportunity** (Environmental Context and Resources) | Pregnant and lactating women, health workers, and community members | Enablement, Environmental restructuring | Develop promotional materials to promote trial activities that are appropriate for the setting, such as radio advertisements and written materials provided at health facilities. Consider how the physical environment where recruitment takes place can be designed to improve person-centredness. |

(*Continued*)

**Table 3.** (Continued)

| Findings | Review findings | COM-B and TDF mapping | Actor | Potential intervention type based on BCW | Examples of how implementation strategies could be operationalised or actual implementation strategies used |
|---|---|---|---|---|---|
| 17 | **Inadequate resources** | **Physical Opportunity** (Environmental Context and Resources) | Study investigators, research staff, and health workers | Enablement, Environmental restructuring | Engage with health workers in the study health facilities to understand any barriers to recruitment given their context. Consider whether it is more appropriate to engage with research staff who are not employed by the health facility for research tasks, such as a research midwife. Ensure that research staffs are compensated fairly for their roles and that trial activities do not interfere with clinical responsibilities. |
| 18 | **Engaging health workers in trials** | **Psychological Capability** (Knowledge, Cognitive/ Interpersonal Skills); **Reflective Motivation** (Beliefs about Consequences, Social/ Professional Role and Identity, Optimism) | Health workers and research staff | Training, Education, Enablement, Persuasion | Depending on the context, health workers may benefit from general training about the utility of trials in maternal health, and how trials relate to evidence-based practice, clinical protocols, and guidelines. Either during formative research before the trial or when training health workers on trial activities, provide ample time for discussion to understand potential barriers to trial implementation and engagement in their setting to assuage fears and ensure benefits are understood. |
| 19 | **Research staff's emotional orientations towards clinical trials** | **Reflective Motivation** (Goals, Beliefs about Capabilities); **Physical Opportunity** (Environmental Context and Resources); **Social Opportunity** (Social Influences) | Research staff | Enablement, Incentivisation, Education, Environmental restructuring | Set feasible, achievable, and clear goals for trial recruitment for each site. Provide recognition and appreciation for research staff to celebrate milestones, achievements, and successful recruitment outcomes. Provide ongoing training to strengthen skills of research staff, offer opportunities for professional development and career growth, and ensure they feel confident and well-equipped for their roles. Consider if establishing performance-based bonuses or rewards are appropriate and could enhance recruitment without negative impacts. Ensure research staff understand the big picture of the significance of their work in advancing knowledge and the broader mission or vision of an organisation or community. |
| 20 | **Women-centred approach encourages participation** | **Psychological Capability** (Knowledge, Memory, Attention, and Decision process); **Reflective Motivation** (Beliefs about Consequences); **Automatic Motivation** (Emotion); **Physical Opportunity** (Environmental Context and Resources) | Pregnant and lactating women, research staff and health workers | Education, Training, Enablement, Environmental restructuring | See finding 1.2. Also consider the most appropriate timing to approach women for trial recruitment, to align and not conflict with timing around their own clinical care. |

*(Continued)*

**Table 3.** (Continued)

| Findings | Review findings | COM-B and TDF mapping | Actor | Potential intervention type based on BCW | Examples of how implementation strategies could be operationalised or actual implementation strategies used |
|---|---|---|---|---|---|
| 21 | **Recruitment for intrapartum research** | **Psychological Capability** (Knowledge, Memory, Attention, and Decision process); **Reflective Motivation** (Beliefs about Consequences); **Automatic Motivation** (Emotion); **Physical Opportunity** (Environmental Context and Resources) | Pregnant and lactating women, partners, research staff, and health workers | Education, Training, Enablement, Environmental restructuring | Timing of intervening for women's participation in research addressing intrapartum interventions is important. If trial information is given to women too early during antenatal care, it may be forgotten during labour. Some women may be distressed in labour, but many women are comfortable in early labour and if they have adequate pain management. Seek input from patient advocates or women's groups, and consider if it is feasible to have a multi-staged approach to providing information about trial participation. For example, women could be sensitised to ongoing trials during antenatal care, with specific recruitment details provided late in pregnancy or when admitted for birth. |
| 22 | **Factors affecting motivation of study investigators** | **Psychological Capability** (Knowledge); **Reflective Motivation** (Goals, Beliefs about Consequences, Social/Professional Role and Identity); **Social Opportunity** (Social Influences) | Study investigators | Education, Persuasion, Incentivisation, Enablement | Work towards creating a research-friendly environment within health facilities. Engage with hospital leadership to promote buy-in, and create research committees. Provide dedicated research space within health facilities (offices, meeting rooms, labs), and invest in research infrastructure (data storage, research information systems). Hire and train research support staff (e.g., research midwives), and provide continuous learning and professional development about conducting research. Where possible, allocate budget for research activities, including small grants, research fellowships, or internal funding for pilot projects. Encourage collaboration between hospital departments, academic institutions, and other research entities, and foster networking opportunities. Hold regular research seminars and journal clubs to promote knowledge exchange. Develop a formal research mentorship scheme for students, trainees, and junior staff. Establish awards or recognition for research achievements. Collaborate with organisations in the facility catchment area to ensure women and community members are involved in research, and research is responsive and relevant to community needs. |

*(Continued)*

**Table 3.** (Continued)

| Findings | Review findings | COM-B and TDF mapping | Actor | Potential intervention type based on BCW | Examples of how implementation strategies could be operationalised or actual implementation strategies used |
|---|---|---|---|---|---|
| 23 | **Challenges in gaining ethical approvals for trials with pregnant women** | **Reflective Motivation** (Optimism [Pessimism], Belief about Consequences); **Physical Opportunity** (Environmental Context and Resources) | Study investigators, ethics committee members, and regulators | Education, Persuasion, Enablement, Environmental restructuring | 1) Educate IRB committee members and regulators about the health consequences of excluding pregnant women from research and the opportunities and approaches for estimating and monitoring risks associated with trial inclusion. 2) Provide training on developing a shared institutional commitment to safe and responsible inclusion of pregnant women in biomedical research as the standard, and develop a common institutional understanding of regulatory guidelines and associated documentation such as standard operating procedures, worksheets, and checklists to facilitate consistency in guideline application by institutional ethics committees and researchers. 3) Promote collaboration, discussion, and consensus-building with ethics committee members in locations where the trials are initiated and implemented, colleagues across different specialties and geographical locations and community representatives through protocol development and study design stages to determine mutually acceptable approaches to trial design. |
| 24 | **Role of funders** | **Reflective Motivation** (Belief about Consequences); **Physical Opportunity** (Environmental Context and Resources) | Funders and research staff | Education, Persuasion, Enablement, Incentivisation, Environmental Restructuring | Awareness-raising for funders about the importance of funding trials with pregnant and lactating women is urgently needed. This should include a clear discussion that if the research is not conducted with these populations, then the responsibility for decision-making about therapeutics is left to health workers and pregnant women: this is an unfair transfer of responsibility. Educational materials could be developed as core funder resources for "how to include pregnant and lactating women" in trials conducted by researchers receiving funds from a given funder. |

levels associated with different stakeholder groups. Which actions are appropriate for a given context should therefore be discussed, prioritised, and adapted to a particular setting.

Some of these strategies may already be in place as part of ethical conduct for trial recruitment, for example, sharing information transparently with potential participants about safety, risks, benefits, and side effects of the trial intervention (BCW intervention type: education). Given pregnant and lactating women's concerns around risks of the intervention, such strategies can be enhanced through personalised discussions about how the intervention relates to women's personal and clinical circumstances, for example, using a decision-aid tool (BCW intervention type: enablement). Developing clear and context-specific ways to explain study design features in plain language, and involvement of trusted sources (such as health workers), to communicate trial information can aid the decision-making process. Engaging with patient advocates and women's groups and conducting formative research with potential participants to receive feedback on acceptability of trial components can streamline trial procedures and enhance acceptability and contextual alignment. Considerations should include how societal and gender norms, and gender roles impact various aspects of participation.

Given potential concerns among health workers regarding safety of interventions during pregnancy, providing access to credible resources on the risks, benefits and potential side-effects of the product being trialled, and elaborating on the trial rationale, potential benefits, and where the trial fits into existing evidence can help address fear and uncertainty regarding intervention safety (BCW intervention type: education, training).

At the health systems-level, strategies include creating a research-friendly environment within health facilities. In addition to promoting buy-in from hospital leadership, this would include infrastructural enhancements such as creating research spaces within health facilities (e.g., offices, meeting rooms, labs, data storage, research information systems), and hiring and training research support staff (e.g., research midwives), among other aspects.

Strategies to promote alignment between study investigators and ethics committee members include: educating ethics committee members about the health consequences of excluding pregnant women from research, and useful approaches for monitoring and managing risks associated with trial inclusion (BCW intervention type: education); developing a shared institutional commitment to inclusion of pregnant women research as the standard, and developing a common understanding of regulatory guidelines and associated documentation such as standard operating procedures, worksheets, and checklists to facilitate consistency in guideline application by institutional ethics committees and researchers.

## Discussion

This review provides a comprehensive overview of the range of factors affecting the participation of pregnant and lactating women in clinical trials across the research ecosystem. At the upstream levels, we identified barriers arising from limited interest of funders to invest in clinical trials with pregnant and lactating women, and reluctance of ethics committees to approve protocols due to potential for risks, particularly to fetal health. Factors at the interface between health systems and communities included developing trusting and reciprocal relationships among community members, research staff, and healthcare workers, and taking a woman-centred approach to recruitment. For women, determining the risk-benefit ratio of participation, trust (or lack thereof) in medicine and research, the potential to access high-quality care through trial participation, and altruistic motivations were key factors. Incorporating a gender lens to the data, we found that participation was impacted by gender relations of power sustained by gender norms, gender role expectations of women as mothers and caregivers, and mixed opinions regarding bodily and decisional autonomy during pregnancy.

Our findings on factors influencing pregnant women's decisions regarding participation are aligned with those identified by Van der Zande and colleagues [98] who found that the potential for personal benefits alongside altruistic motivations were crucial drivers, while participation burdens, risks, and mistrust in research were key barriers to participation. Some of these findings, such as the role of altruism and potential for personal benefit, concerns about randomisation and other study design features, burdensome trial procedures, fears associated with taking an experimental therapy, and health worker attitudes towards trials are also consistent with the broader literature on factors associated with trial participation [99–101]. Across the findings, women and research staff emphasised the importance of a woman-centred approach to trial recruitment, with careful consideration of women's individual clinical and personal circumstances, transparency in information, and support for informed and unhurried decision-making. These aspects were found to be challenging to navigate in intrapartum trials, given the timing of recruitment coinciding with birth, often in the context of impending or ongoing complications. For example, a recent analysis of uterotonic trials for prevention of postpartum haemorrhage found considerable variability between trials in the timing of informed consent—most obtained consent during labour, with a minority in the antenatal period [102]. Our findings suggest that women prefer consent in the antenatal period to optimise informed and unhurried decision-making. However, there are ethical concerns about seeking antenatal consent as it may exclude participation of women who do not regularly access antenatal care [102]. Indeed, the informed consent process in intrapartum trials is an issue of current debate and ethical interest [103], and more empirical work is needed to understand women's preferences and needs to optimise informed decision-making.

We found that healthcare workers' engagement was crucial in recruiting women as they play a vital role in bridging communication between potential participants and research staff. Many studies reported that women relied on health workers advice in making decisions about participation. Health workers in turn encouraged or discouraged participation based on their own attitudes towards clinical research in pregnancy and knowledge about or personal experience using the therapy under investigation. Given the roles of trust and power in women's decision-making processes, it is important to promote transparent and open communication between women and health workers regarding trials, and their associated risks and benefits [104,105]. It is also important to clarify differences between clinical trial and regular clinical care to minimise the potential for therapeutic misconceptions, the consequences of which could lead to the eroding of trust in the medical system, affecting future health-seeking behaviour.

The complicated issue of autonomy in decision-making during pregnancy was raised by multiple stakeholders. Many women discussed trial participation with their partners and other family members but considered the final decision to be their own. In some settings, usually in the context of rigid gender norms, women required partners' permission to participate; if violated, this could result in the threat of violence or marital discord. Separately, the imposition of a paternal consent requirement was viewed as a significant barrier for women who were in unstable relationships, unmarried, or wanted to exercise fully autonomous decision-making. Widmer and colleagues [102] argue that it is the role of research staff to guarantee and protect women's autonomy. We found that women's decisional autonomy was impacted by intimate partner relationship dynamics, and wider sociocultural and gender norms that required nuanced understandings of the context and multistakeholder engagement to create an enabling environment for women to exercise choice.

We also identified barriers experienced by researchers, ethics committees, and funders of clinical trials. Study investigators had trouble obtaining ethical approval as ethics committees have mixed perspectives on the inclusion of pregnant and lactating women in trials,

particularly in the absence of clear guidelines. In line with previously reported upstream barriers [16, 23], limited interest in funding clinical trials with pregnant and lactating women due to potential risks, high liability, and reputational consequences also inhibits the implementation of trials. These findings demonstrate a need to develop holistic strategies addressing barriers experienced by stakeholders operating at the upstream levels of clinical research.

The TDF and COM-B mapping in our review (Table 3) can be used by study investigators, research staff, health workers, ethics committees, and funders to inform the development of implementation strategies to address barriers to pregnant and lactating women's participation in clinical trials. Formative research to identify specific barriers and facilitators in specific settings and contexts is a recommended starting point before developing appropriate strategies.

A limitation is that we did not include grey literature, which may have expanded the types of evidence and/or contexts of the review. However, our search strategies yielded high coverage of published literature. The studies included in the review had good coverage of countries from the African region, but sparse representation of countries from Latin America, and no representation of countries in the Eastern Mediterranean or South-East Asian regions. A growing number of trials addressing maternal and perinatal health are being implemented in these settings [106], calling for significantly greater focus in formative and process evaluation research with pregnant and lactating women and people, family members, health workers, local researchers, and ethics committee members to understand context-specific motivations for and concerns regarding conduct of and participation in research during pregnancy and lactation. The AIM-Gender project [107] aims to address this limitation through qualitative research on the topic in India and Nigeria—2 countries that together account for 37% of global maternal deaths [13]. Future work must also consider inclusion of pregnancy-capable transgender and nonbinary people, as knowledge gaps regarding factors affecting their participation in pregnancy and lactation clinical research are particularly pronounced. We also draw attention to 2 relevant reviews on factors affecting participation of racial and ethnically marginalised populations in pregnancy and lactation research, a related topic that was beyond the scope of this review [108,109].

Our review builds on previous work [98] by examining the full range of factors and perspectives of multiple stakeholders operating at the upstream and downstream levels of the research ecosystem. We optimised the available data by including qualitative, quantitative, and mixed-methods primary research. We applied the GRADE-CERQual approach to assess confidence in each finding, i.e., the extent to which the finding adequately represented the phenomenon of interest [32,33]. These assessments have important practical implications for increasing the applicability and usability of these findings by stakeholders seeking to enhance research and development in maternal health. This review additionally integrates the use of behavioural frameworks [24,25] to propose a theory-informed set of behaviour change interventions to address factors affecting clinical trial participation among pregnant and lactating women.

## Supporting information

**S1 Appendix. Preferred reporting items for systematic reviews and meta-analyses (PRISMA) reporting checklist.**
(DOCX)

**S2 Appendix. Enhancing transparency in reporting the synthesis of qualitative research (ENTREQ) reporting checklist.**
(DOCX)

**S3 Appendix. Search strategies.**
(DOCX)

**S4 Appendix. GRADE-CERQual evidence profile.**
(DOCX)

**S5 Appendix. Summaries of quantitative findings.**
(DOCX)

**S6 Appendix. Characteristics of included papers.**
(DOCX)

**S7 Appendix. Critical appraisal.**
(DOCX)

## Acknowledgments

We are grateful to Alessandra Fleurent at Concept Foundation for her assistance with verifying the accuracy of the translated French paper included in this review.

## Author Contributions

**Conceptualization:** A. Metin Gülmezoglu, Meghan A. Bohren.

**Data curation:** Mridula Shankar, Patrick Condron, Meghan A. Bohren.

**Formal analysis:** Mridula Shankar, Alya Hazfiarini, Rana Islamiah Zahroh, Annie R. A. McDougall, Anne Ammerdorffer, Meghan A. Bohren.

**Funding acquisition:** A. Metin Gülmezoglu, Meghan A. Bohren.

**Methodology:** Mridula Shankar, Alya Hazfiarini, Rana Islamiah Zahroh, Joshua P. Vogel, Annie R. A. McDougall, Patrick Condron, Shivaprasad S. Goudar, Yeshita V. Pujar, Manjunath S. Somannavar, Umesh Charantimath, Anne Ammerdorffer, Sara Rushwan, A. Metin Gülmezoglu, Meghan A. Bohren.

**Project administration:** Mridula Shankar.

**Supervision:** Meghan A. Bohren.

**Validation:** Mridula Shankar, Alya Hazfiarini, Meghan A. Bohren.

**Visualization:** Mridula Shankar, Alya Hazfiarini.

**Writing – original draft:** Mridula Shankar, Alya Hazfiarini, Meghan A. Bohren.

**Writing – review & editing:** Mridula Shankar, Alya Hazfiarini, Rana Islamiah Zahroh, Joshua P. Vogel, Annie R. A. McDougall, Patrick Condron, Shivaprasad S. Goudar, Yeshita V. Pujar, Manjunath S. Somannavar, Umesh Charantimath, Anne Ammerdorffer, Sara Rushwan, A. Metin Gülmezoglu, Meghan A. Bohren.

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
