## [Editor Report · Decision Letter 0]

21 Dec 2023

Dear Dr Shankar, 

Thank you for submitting your manuscript entitled "Factors influencing the participation of pregnant and lactating women in clinical trials: a mixed-methods systematic review" for consideration by PLOS Medicine.

Your manuscript has now been evaluated by the PLOS Medicine editorial staff as well as by an academic editor with relevant expertise and I am writing to let you know that we would like to send your submission out for external peer review.

Ordinarily we request that you please re-submit your manuscript within two working days, i.e. by Dec 25 2023 11:59PM, but we appreciate that there may be delay due to the holiday period which we are not concerned about.

Kind regards,

Pippa

Philippa Dodd, MBBS MRCP PhD

PLOS Medicine

pdodd@plos.org

---

## [Decision Letter · Decision Letter 1]

15 Feb 2024

Dear Dr. Shankar,

Many thanks for submitting your manuscript "Factors influencing the participation of pregnant and lactating women in clinical trials: a mixed-methods systematic review (PMEDICINE-D-23-03776R1) to PLOS Medicine. The paper has been reviewed by two subject experts and a statistician; their comments are included below and can also be accessed here: [LINK]

As you will see, the reviewers were very positive about the paper and raised nothing by way of concern but ask for clarifications and some additional considerations as constructive suggestions for improvement. After discussing the paper with the editorial team and an academic editor with relevant expertise, I’m pleased to invite you to revise the paper in response to the reviewers’ comments. We plan to send the revised paper to some of all of the original reviewers*, and of course we cannot provide any guarantees at this stage regarding publication.

When you upload your revision, please include a point-by-point response that addresses all of the reviewer and editorial points, indicating the changes made in the manuscript and either an excerpt of the revised text or the location (eg: page and line number) where each change can be found. Please submit a clean version of the paper as the main article file and a version with changes marked should as a marked-up manuscript. Please also check the guidelines for revised papers at http://journals.plos.org/plosmedicine/s/revising-your-manuscript for any that apply to your paper.

We ask that you submit your revision by March 7th 2024. However, if this deadline is not feasible, please contact me by email, and we can discuss a suitable alternative.

Please don’t hesitate to contact me directly with any questions (pdodd@plos.org). If you reply directly to this message, please be sure to ‘Reply All’ so your message comes directly to my inbox.

Kind regards,

Pippa

Philippa Dodd, MBBS MRCP PhD

PLOS Medicine

plosmedicine.org

pdodd@plos.org

*Please note: If your article is accepted, you may have the opportunity to make the peer review history publicly available. The record will include editor decision letters (with reviews) and your responses to reviewer comments. If eligible, we will contact you to opt in or out.

Editorial comments:

1) The editorial team are in agreement with the reviewers that your manuscript was very considerately designed and reported and that it could make a valuable contribution to the field. 

2) We thank you for reporting your study in accordance with PRISMA guidelines and for including the checklist as supporting information. PLOS Medicine requires that systematic reviews are updated to within 6 months of an anticipated publication date. We appreciate the complexity of your particular study but please consider this point as part of your revision.

Comments from the reviewers:

Reviewer #1: See attachment

Michael Dewey

Reviewer #2: In this manuscript, the authors present a sweeping and comprehensive systematic review of qualitative and quantitative studies addressing the participation of pregnant and lactating women in clinical trials. This mixed methods review is very well done and contains excellent information. A few comments and questions follow.

1. The introduction is excellent and provides a sound basis for the study. This could be an introduction given to policymakers and stakeholders in its current form.

2. Page 5 first paragraph listing the exclusion on letter F-why were studies excluded if they looked at engagement in observational research? Given that the entire focus of the paper was to analyze and synthesize facilitators and barriers to engaging pregnant women in research, how is it justifiable to exclude inclusion in observational studies?

3. The findings overall were things that have been talked about for years. One of the key things that was exciting about this manuscript was the potential to put together a list of accomplishable behavior change strategies. However, the manuscript does not really do this in an effective way. Table 2 maps the findings to behavior change interventions, but still has 24 different things to accomplish. This becomes overwhelming and demotivating for investigators, participants, and governing and regulatory bodies. It would have been extremely helpful for a shorter prioritized list to be developed for change strategies that could be implemented quickly.

4. Many of the 24 different findings are somewhat related and could have been collapsed together.

Reviewer #3: I have a read the manuscript with great interest. As the authors say, their systematic review is an elaboration of a previous systematic review performed by Van der Zande et al (2018) that looked at women's reasons for participation in clinical research. Since the authors took a more comprehensive approach (a broader group of stakeholders and a broader set of studies), the result is impressive: 24 well-considered key findings. 

I have a few suggestions.

 A well-known problem is that studies that measure the willingness to participate are often based on retrospective data and lead to recall bias. Could data be added about the type of studies that was found?

 Could data be added about the type of clinical research that was studied? As the authors acknowledge in the introduction there is a scarcity on pregnancy specific medications. Can more be said on the type of studies found? Obstetric, non-obstetric, pharmacological, non-pharmacological?

[LINK]

Comments from the academic editor:

I think this is a thorough and carefully conducted study and the conclusions make a lot of sense. None of the reviews indicate any major flaw and it feels like a rigorous piece of work. I agree with your decision to send it out for full review and to revise it as this point.

1. Please upload any figures associated with your paper as individual TIF or EPS files with 300dpi resolution at resubmission; please read our figure guidelines for more information on our requirements: http://journals.plos.org/plosmedicine/s/figures. While revising your submission, please upload your figure files to the PACE digital diagnostic tool, https://pacev2.apexcovantage.com/. PACE helps ensure that figures meet PLOS requirements. To use PACE, you must first register as a user. Then, login and navigate to the UPLOAD tab, where you will find detailed instructions on how to use the tool. If you encounter any issues or have any questions when using PACE, please email us at PLOSMedicine@plos.org.

To submit your revised manuscript, please use the following link:

---

## [Decision Letter · Decision Letter 2]

10 Apr 2024

Dear Dr. Shankar,

Thank you very much for re-submitting your manuscript "Factors influencing the participation of pregnant and lactating women in clinical trials: a mixed-methods systematic review" (PMEDICINE-D-23-03776R2) for review by PLOS Medicine.

I have discussed the paper with my colleagues and the academic editor and it was also seen again by 2 reviewers. I am pleased to say that provided the remaining editorial and production issues are dealt with we are planning to accept the paper for publication in the journal.

[LINK]

We look forward to receiving the revised manuscript by Apr 17 2024 11:59PM.   

Kind regards,

Pippa

Philippa Dodd, MBBS MRCP PhD

PLOS Medicine

plosmedicine.org

pdodd@plos.org

Requests from Editors:

GENERAL

Thank you for your very detailed and considered responses to previous comments. Please see below for further minor comments which we require you address prior to publication.

These requests pertain to specific content and formatting requirements some may have already been incorporated into the manuscript and some may not be directly relevant but please review the complete list and all that all items are included as relevant.

ABSTRACT

In the last sentence of the Abstract Methods and Findings section, please (briefly) describe the main limitation(s) of the study's methodology.

AUTHOR SUMMARY

Thank you for including an author summary which reads very nicely. 

In the final bullet point of ‘What Do These Findings Mean?’, please describe the main limitations of the study in non-technical language.

We appreciated the inclusive reference to ‘lactating women and people’ in the opening bullet points but the same isn’t continued throughout. I noted your clarification regarding terminology Vs applicability to gender diverse populations (methods, line 182 onwards) which is very helpful to the reader and to me explains the aforementioned. Perhaps including a similar statement as part of the limitations (as requested above) would mitigate against any confusion regarding your use of terminology in the opening bullet points and the description of the research outcomes latterly. 

INTRODUCTION

As above, please consistently refer to ‘women and people’, (or populations) line 124, for example. Please check carefully throughout the complete manuscript (including the supporting information) and revise as necessary.

TABLES

Please ensure that all tables (including those in the supporting information) are affiliated to an appropriate title and caption which clearly describes their content without the need to refer to the text.

DISCUSSION

Please present and organize the Discussion as follows: a short, clear summary of the article's findings; what the study adds to existing research and where and why the results may differ from previous research; strengths and limitations of the study; implications and next steps for research, clinical practice, and/or public policy; one-paragraph conclusion.

REFERENCES

For in-text reference callouts please place citations in square parentheses separate by commas but no additional spaces. For example, [1,3,6] or [1-3]. Please check and amend throughout all sub-sections of the manuscript and supporting files.

In the bibliography please ensure that you list up to but no more than 6 author names followed by et al.

For all web references please ensure you include an, ‘Accessed [date].’

Journal name abbreviations should be those listed in the National Center for Biotechnology Information (NCBI) databases.

SUPPORTING INFORMATION

Please ensure to cite your Supporting Information as outlined here: https://journals.plos.org/plosmedicine/s/supporting-information

In the published article, supporting information files are accessed only through a hyperlink attached to the captions. For this reason, you must list captions at the end of your manuscript file. You may include a caption within the supporting information file itself, as long as that caption is also provided in the manuscript file. Do not submit a separate caption file.

Please ensure to upload a clean (as well a tracked) version of supporting information files.

SOCIAL MEDIA

To help us extend the reach of your research, please detail any X (formerly Twitter) handles you wish to be included when we tweet this paper (including your own, your coauthors’, your institution, funder, or lab) in the manuscript submission form when you re-submit the manuscript.

Comments from Reviewers:

Reviewer #1: The authors have answered my queries and addressed my other points.

Michael Dewey

Reviewer #3: Thanks for addressing the comments. I had asked for table 1 and table S6, we I could not find in the original resubmission. The additions in table 1 are helpful. And table S6 appears to be a typo (it should be Appendix S6). If these small comments are addressed than I have no further questions.

[LINK]

---

## [Editor Report · Decision Letter 3]

19 Apr 2024

Dear Dr Shankar, 

On behalf of my colleagues and the Academic Editor, Professor Gordon Smith, I am pleased to inform you that we have agreed to publish your manuscript "Factors influencing the participation of pregnant and lactating women in clinical trials: a mixed-methods systematic review" (PMEDICINE-D-23-03776R3) in PLOS Medicine.

PRESS

Thank you again for submitting to PLOS Medicine, it has been a pleasure handling your manuscript. We look forward to publishing your paper. 

Kind regards,

Pippa

Philippa Dodd, MBBS MRCP PhD  

PLOS Medicine

pdodd@plos.org